# Mesocrystal growth through oriented sliding and attachment of nanoplates

Xiaoxu Li [1], Tuan A. Ho[2], Honghu Zhang [3], Lili Liu [1], Ruipeng Li[3], Ping Chen[1], Mark E. Bowden [4], Sebastian T. Mergelsberg [1], Hongyou Fan [2], James J. De Yoreo [1], Carolyn I. Pearce [5], Kevin M. Rosso [1] ✉ & Xin Zhang [1] ✉

Oriented attachment is a critical, yet poorly understood, crystal growth pathway based on the self-assembly of nanocrystals. During oriented attachment, solvent-separated particles align and coalesce through forces that enable precise rotation and translation. While prior studies emphasized intragap forces driving crystallographic alignment, the forces enabling uniform stacking and superlattice formation remain unclear. Here, we demonstrate how macroscopic gibbsite mesocrystals emerge from nanoplates guided into staggered positions by directional sliding. Electron microscopy and X-ray scattering reveal the monoclinic superlattice structure, based on nanoplate stacking with a uniform ≈50° stagger along the gibbsite [010] direction. In situ liquid-cell TEM captures preferential sliding along the gibbsite [010] direction, decelerating with increasing particle overlap. Molecular dynamics simulations reveal that this staggered arrangement corresponds to a global free-energy minimum, rather than full alignment. The simulations also confirm that sliding along the [010] direction is energetically favored and provide insight into the role of interfacial water in achieving long-range ordered assemblies. These insights highlight the energy landscape's role in oriented attachment, with implications for material synthesis and hierarchical structures in nature.

Particle-mediated crystallization is widespread in geochemical, biomineral, and synthetic material systems[1]. Oriented attachment (OA) drives nanocrystals to self-assemble into larger crystals via interparticle forces based on relative crystallographic orientation[2]. This process enables a disorder-to-order transition of nanoparticle suspensions, forming superlattices or mesocrystals with shared atomic-scale orientations[3]. Distinct from classical monomer-by-monomer addition, OA enables the fusion of coaligned crystals into larger structures[1], providing opportunities to construct complex, hierarchical materials with desired properties in bioscience and materials

science applications[4]. A fundamental understanding of particle-mediated crystallization by OA is also essential for properly interpreting the occurrences and geological implications of natural hierarchical mineral structures, such as those found in ores (e.g., hematite[5]) and biominerals[6].

Recent breakthroughs in in situ observations have deepened our understanding of OA in liquids. Liquid-cell transmission electron microscopy (LCTEM) studies revealed that iron hydroxide nanocrystals undergo repeated incidental contact and detachment by diffusion and rotation until sufficient alignment enables an attractive jump-to contact

[1]Physical and Computational Sciences Directorate, Pacific Northwest National Laboratory, Richland, WA, USA. [2]Geochemistry Department, Sandia National Laboratories, Albuquerque, NM, USA. [3]National Synchrotron Light Source-II, Brookhaven National Laboratory, Upton, NY, USA. [4]Institute for Integrated Catalysis, Pacific Northwest National Laboratory, Richland, WA, USA. [5]Energy and Environment Directorate, Pacific Northwest National Laboratory, Richland, WA, USA. ✉e-mail: kevin.rosso@pnnl.gov; xin.zhang@pnnl.gov

and fusion[7]. In contrast, ligand-coated gold nanoparticles align through overlapping ligands before contact[8]. For ZnO nanoparticles, initial co-alignment forces span nanometer-scale gaps, followed by strong attractions that drive contact and fusion[9]. These seminal observations highlight the importance of the strength of both torsional and translational forces spanning the nanometer-scale gap between particles in the solvent-separated state. Advanced experimental techniques, including atomic force microscopy-based methods, have enabled direct measurements of these forces, including those for TiO$_2$[2,10], ZnO[11], and mica[12], as a function of azimuthal angle and separation distance. Molecular simulations reveal how these forces arise from a complex interplay of directional hydrogen bonding networks, van der Waals forces, dipolar interactions, and ion correlation forces[13].

Despite the advances detailed above, most OA research focused on the attractive or torsional forces enabling atomic alignment, overlooking how docking crystallites achieve uniform stacking and aggregate-scale alignment in mesocrystals. Conceptually, to achieve this whole-particle alignment points to the importance of forces that govern interparticle translation, or 'sliding', while still in the solvent-separated state. When such particles are pre-aligned to have commensurate lattices by intragap forces, this sliding motion can have a directional dependence. Particle morphology is likely an essential additional aspect. For example, particularly for two-dimensional nanocrystals, particle edges may bear a special surface charge affecting the collective electrostatic interaction between unaligned particles. Crystal shape anisotropy can also influence particle diffusion through viscosity effects in the surrounding medium[14]. Although the self-assembly of two-dimensional materials has been widely reported[15–18], the explanation of their assembly into mesoscale structures still relies on simple constructs such as the hard-sphere[19] or coarse-grained models[16]. Few studies have directly linked the structure of mesocrystals with the directional interactions between particles at the atomic level. Thus, by comparison to studies on attractive forces and the achievement of co-alignment, little attention has been directed toward these lateral forces governing the 'sliding into place' of particles, an omission that remains key for achieving a comprehensive understanding of the OA process.

Beyond OA, lateral translation forces between particles are crucial in diverse fields, including the soil shear strength and stability[20], the mechanical properties of materials in powder metallurgy[21], and the flow behavior of granular fluid mixtures[22]. Moreover, understanding these forces is fundamental to the study of friction, wear, and lubrication, where atomic-level insights can aid in designing advanced lubricants to reduce friction, minimize wear, and extend the longevity of mechanical components[23]. Despite their broad significance, the thermodynamic driving forces and free-energy landscape during lateral translation, along with their relationship to azimuthal alignment, interfacial solvent structure, and atomistic mismatch during translation, remain poorly understood.

Here, we report an OA system of gibbsite nanoplates with uniform hexagonal shape whose specific stacking behavior in water is central to their self-assembly into mesocrystals. The results highlight the importance of translational, torsional, and sliding forces working in concert to yield three-dimensional mesocrystals of macroscopic proportions. Additionally, we underscore the pivotal role of the underlying energy-structure relationship governing both the interparticle sliding motion and the resulting structure of the gibbsite mesocrystals. Our findings expand the picture of the forces and motions that produce OA in two-dimensional material systems, offering fundamental insights that could ultimately enable predictive models for this process.

## Results and discussion
### The formation and structure of gibbsite mesocrystals
In the crystal structure of gibbsite, aluminum atoms occupy 2/3 of the octahedral interstices between close-packed layers of hydroxyl (OH) groups, forming Al-centered octahedral sheets that are interconnected by hydrogen bonds. Using a two-step hydrothermal method without surfactants[24], we synthesized hexagonal gibbsite nanoplates with a uniform morphology. The nanoplates measured $92.7 \pm 5.5$ nm in diameter and $9.5 \pm 2.1$ nm in thickness, featuring two exposed (001) basal planes and six edge surfaces (two (100) and four (110) facets), as confirmed by TEM[24].

The gibbsite nanoplates were well dispersed in deionized (DI) water after 30 minutes of intense ultrasonication, resulting in a dilute suspension (0.81 vol.%). The initial pH of the suspension was ≈5.6. Given that the point of zero charge for gibbsite is typically in the range of 8.7–11.0, the nanoplates were nominally positively charged with a high zeta potential (20–45 mV)[25]. However, the basal planes and edge facets exhibit distinct surface structures and acidities[26]. For the basal plane, ab initio calculation revealed that some OH groups align parallel to the surface, acting only as proton acceptors with low pKa values (≈1.3). Other OH groups have a very high pKa (≈22.0), indicating that dissociation does not occur at pH 5.6. Therefore, the OH groups of the basal planes do not significantly contribute to the acid–base chemistry of gibbsite at pH 5.6. Recent AFM studies have revealed that even the (001) basal surface carries a slight but measurable positive surface charge at pH <6[27], primarily due to ion adsorption at defect sites[28]. Despite this, the basal plane remains weakly charged relative to the edge surfaces, which carry ≡Al(OH$_2$)$^{2+}$ groups with pKa values of 9.0–10.0. As a result, the nanoplates can be viewed as having quasi-neutral basal surfaces and positively charged edges under our experimental conditions (pH 5.6, low ionic strength). In the absence of electrostatic forces between the basal planes of two nanoplates, attractive van der Waals interactions and hydrogen bonds are expected to provide a net attractive global minimum in free energy[29].

The gibbsite nanoplate suspension was left undisturbed in a sealed container at room temperature for six months, resulting in the aggregation and sedimentation of gibbsite nanoplates. At the bottom of the sediment, we collected visibly flake-shaped aggregates ranging in size from a few micrometers to several hundred micrometers. Under polarized-light optical microscopy, domains with identical crystallographic orientations were visible within the aggregates, indicating that the aggregates possessed at least one-dimensional orientational order within each domain (Fig. 1a, b). X-ray diffraction (XRD) pattern of single gibbsite aggregates using a Mo X-ray source (Fig. 1c) shows that the crystallographic orientations of constituent gibbsite nanoplates exhibit co-alignment: although the diffraction spots rotate around the center within 10°–20° mosaicity, the two-dimensional diffraction pattern reveals that the gibbsite nanoplates within the aggregate of several hundred micrometers exhibit three-dimensional orientational order. This result indicates that the as-formed aggregates are gibbsite mesocrystals. Detailed index analysis confirms that the aggregate XRD pattern is a combination of diffraction patterns from different zone axes, including the [010], [130], [100], and [110] zone axes. These zone axes are all parallel to the basal plane of gibbsite but have different crystallographic orientations out of the basal plane. The honeycomb-like crystal structure of the Al(OH)$_3$ sheets in gibbsite exhibits a nominal threefold symmetry axis. Hence, among these zone axes, [010] and [130], or [100] and [110] share very similar diffraction patterns, except for a small offset in distance and angle, since gibbsite has a monoclinic atomic structure ($\beta = 94.54°$). For clarity, only the indexing of the [100] and [010] zone axes is labeled in Fig. 1c. Detailed indexing is shown in the high-resolution synchrotron X-ray scattering data.

Electron microscopy revealed the detailed structure of the gibbsite mesocrystals. To observe the internal structure, we subjected an entire mesocrystal to short-time ultrasonication for fragmentation, followed by characterization using scanning electron microscopy (SEM) and TEM. SEM images (Suppl. Fig. 1) show ≈20 μm mesocrystals. Higher-magnification images reveal that the gibbsite at both the edge

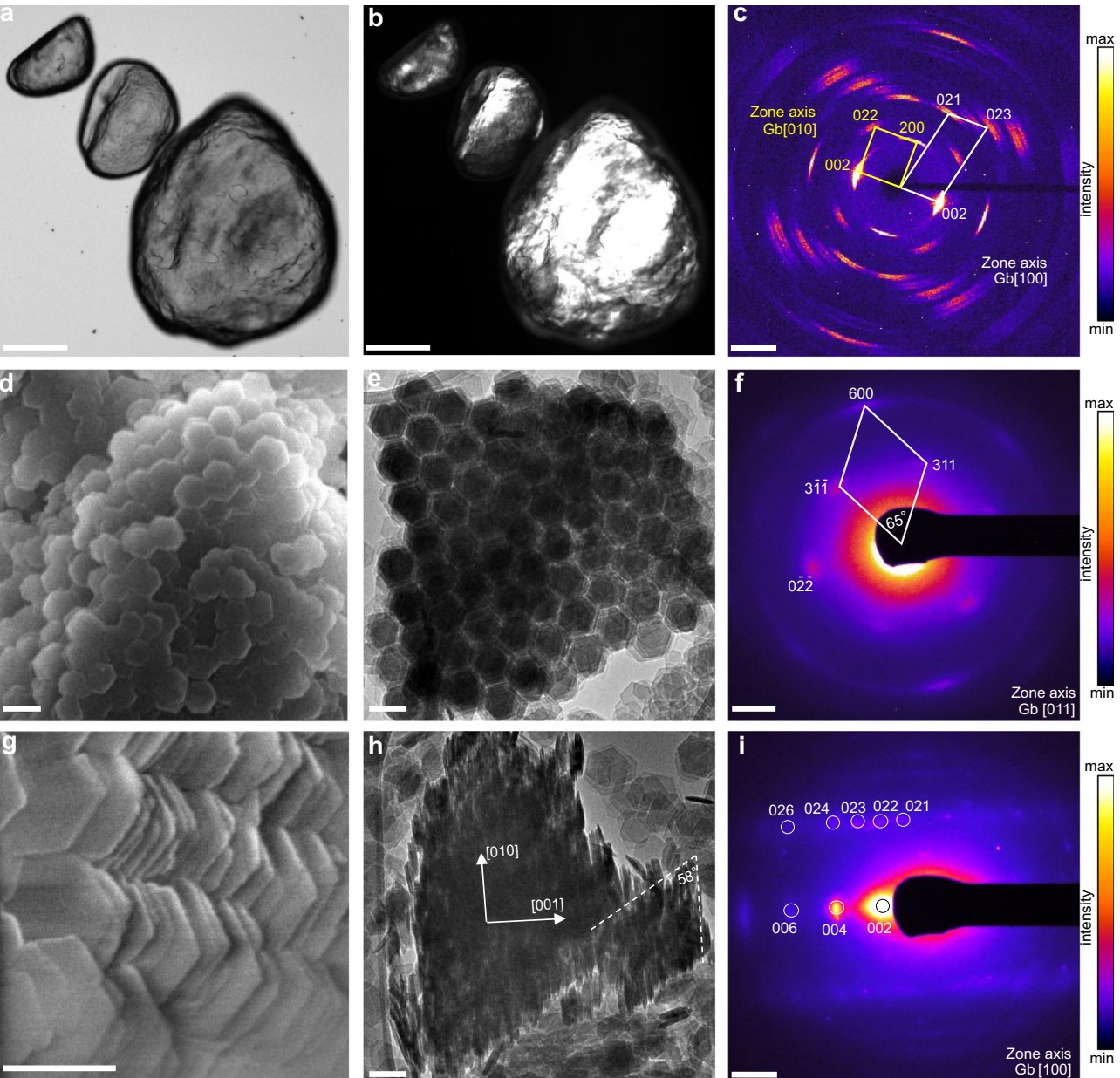

**Fig. 1 | SEM and TEM characterization of the gibbsite mesocrystal precipitated in DI water. a** Normal and **b** polarized-light optical microscopy images. **c** Single-crystal X-ray diffraction pattern showing overlapping zone axes [010] and [100] of gibbsite. **d** SEM image of the mesocrystal viewed from the basal surface. **e** TEM image of the gibbsite mesocrystal from the top view, showing a hexagonal arrangement of gibbsite nanoplate columns. **f** SAED pattern corresponding to (**e**), indexed to the gibbsite [011] zone axis. **g** SEM image showing nanoplates stacked along the basal planes with a lateral stagger. **h** Side-view TEM image showing a 58° angle between the nanoplate basal plane and the stacking axis; the stagger direction corresponds to the [010] direction of gibbsite. **i** SAED pattern corresponding to (**h**), indexed to gibbsite [100]. Scale bars: (**a**, **b**) 100 μm; (**d**, **e**, **g**, **h**) 100 nm; (**c**, **f**, **i**) 2 nm⁻¹. The intensity values in (**c**, **f**, **i**) are plotted on a linear scale.

and top surfaces of the aggregate exhibits a consistent crystallographic orientation, suggesting that the orientational and positional order of the mesocrystals is established and maintained during the aggregation process. Simultaneous observation of XRD patterns (Fig. 1c) with different zone axes likely results from the patchwork of domains with varying crystallographic orientations within larger mesocrystals (tens of micrometers in size).

Figure 1d, e presents the top-view SEM and TEM images of a mesocrystal. Nanoplates are stacked via basal–basal attachment into columns, organized in a hexagonal pattern. Interestingly, the corresponding selected area electron diffraction (SAED, Fig. 1f) of the mesocrystal shows that the zone axis of the gibbsite nanoplates in Fig. 1e is close to the gibbsite [011], rather than [001], with the latter

being the direction normal to the basal plane of the nanoplate. The hexagonal superlattice structure disappears from the TEM images when the electron beam is aligned with the gibbsite [001] zone axis (Suppl. Fig. 2). This indicates adjacent nanoplates displaced along the [010] direction. SEM and TEM images confirm this observation (Fig. 1g, h). The displacement of nanoplates along the [010] direction results in a 58° angle between the basal plane of the nanoplates and the stacking axis (Fig. 1h, i). Further TEM image analysis indicates that the average displacement varies between 6 and 12 nm, corresponding to a range in angle of 58° to 40° between the nanoplate basal plane and the stacking axis (Suppl. Fig. 3).

To resolve the long-range order of the gibbsite mesocrystals and their linkage to the crystallographic orientation of the building blocks,

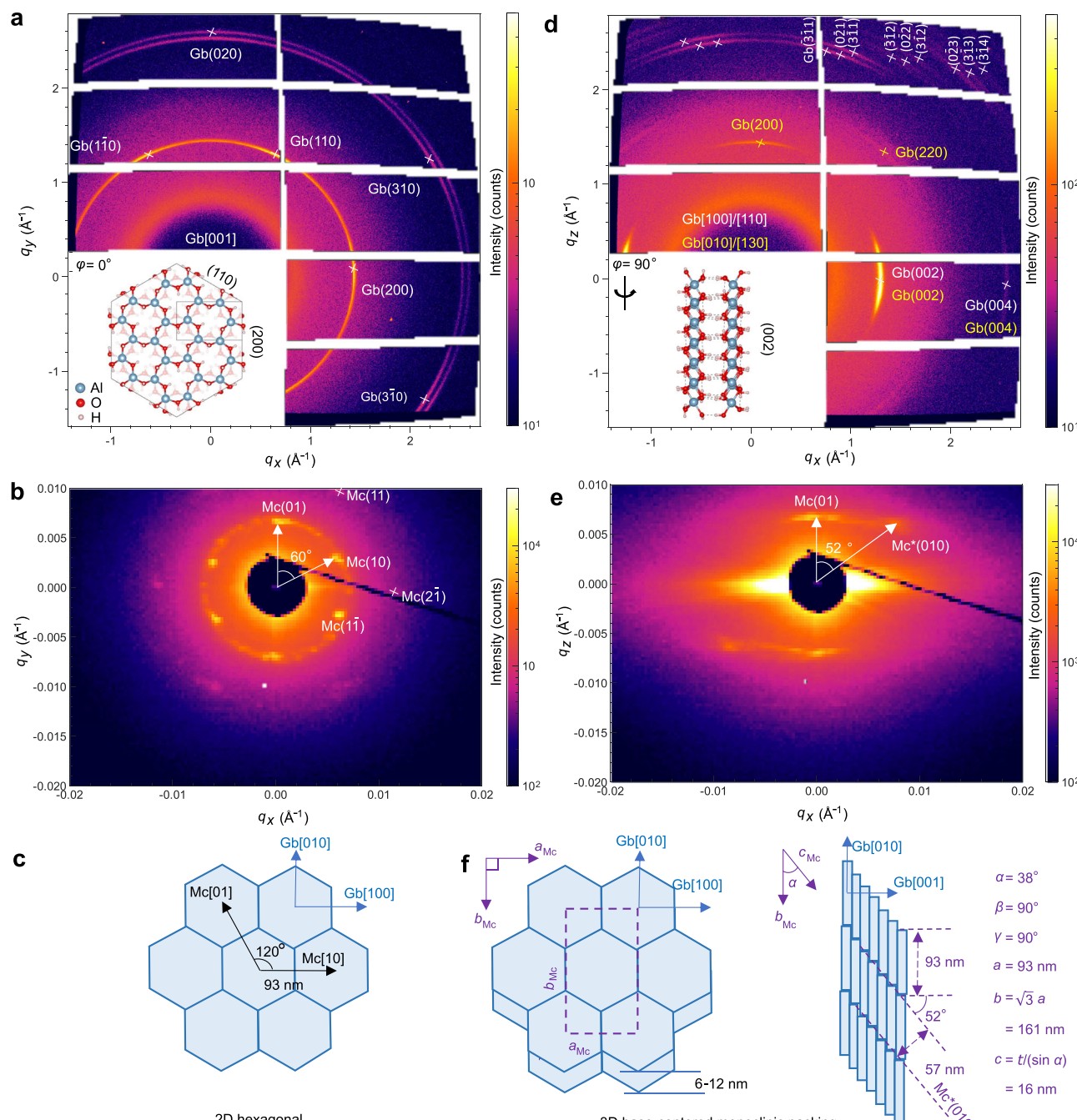

**Fig. 2 | Single-crystal wide- and small-angle X-ray scattering patterns of a single gibbsite mesocrystal.** Gibbsite lattice indices are labeled as "Gb", and mesocrystal indices as "Mc". **a**, **b** WAXS/SAXS patterns with the mesocrystal placed in the *x–y* plane and the X-ray beam oriented perpendicular to the Gb(001) plane. **c** Structure of the mesocrystal viewed along the Gb[001] direction, showing a two-dimensional hexagonal lattice. **d**, **e** WAXS/SAXS patterns with the mesocrystal placed in the *y–z* plane and the X-ray beam parallel to the Gb(001) plane. **f** Structure of mesocrystal shown as a three-dimensional base-centered monoclinic lattice. The intensity values in (**a**, **b**, **d**, **e**) are plotted on a logarithmic scale.

we analyzed a single aggregate with synchrotron wide- and small-angle X-ray scattering (WAXS/SAXS). We denote the gibbsite lattice indices as "Gb" and mesocrystal indices as "Mc".

When the incident beam is aligned close to the Gb[001] axis (Fig. 2a, b), WAXS shows sharp spots on a faint Debye ring, and SAED confirms quasi-single-crystal order with an azimuthal mosaicity of <10° (Suppl. Fig. 2). This angular range matches the potential-of-mean-force minimum for in-plane rotation predicted by Ho et al., which shows that basal–basal interactions in the solvent-separated state favor a relative orientation within ≈10° of perfect alignment[29]. The concurrent SAXS pattern shows a clear sixfold positional symmetry with a nearest-

neighbor distance of 93 nm, corresponding to the edge-to-edge size of individual nanoplates (Fig. 2c). The pattern indexes to a centered rectangular 2D lattice with $a_{Mc}$ = 93 nm and $b_{Mc} = \sqrt{3}\, a_{Mc}$ (Fig. 2f).

After rotating the aggregate by ≈90° (Fig. 2d, e), the WAXS reflections are indexed to zone axes Gb[010]/[130] and Gb[100]/[110], confirming that the beam now lies parallel to the platelet basal planes. Diamond-shaped equatorial streaks in SAXS reveal anisotropic diffuse scattering along Gb[002], consistent with the stacking direction. The absence of Mc*(001) reflections indicates orientational alignment of gibbsite nanoplates along the Gb[002] direction, without clear long-range translational periodicity. In addition to the 2D Mc(01) spots, a

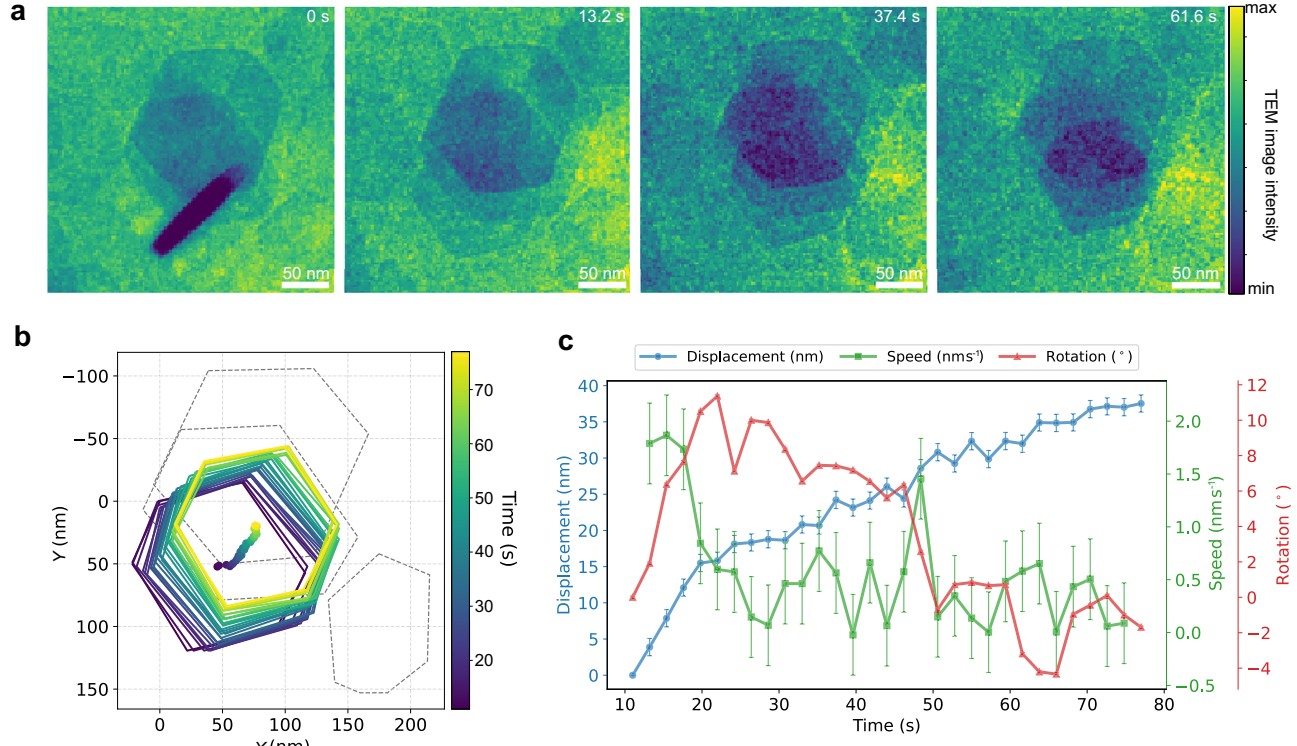

**Fig. 3 | In situ liquid-cell TEM observation of oriented attachment of gibbsite nanoplates from Supplementary Movie 1. a** Selected frames (2.2 s per frame) showing the transition from an initially tilted platelet to an in-plane configuration. The TEM intensity values are plotted on a linear scale. **b** Outlines and centroid trajectory of moving nanoplate after flip ($t = 11.0$ s). Several neighboring particles are outlined with dashed lines. **c** Displacement (blue), instantaneous speed (green), and rotation (red) of the moving nanoplate over time, extracted from the centroid trajectory in (**b**). Error bars denote one standard deviation of the propagated uncertainties: translational displacement and speed uncertainties were estimated from the image resolution (0.6 pixel nm⁻¹), while rotational uncertainties were determined by bootstrap resampling of contour coordinates (200 iterations). Source data of (**b**, **c**) are provided as a Source Data file.

new reflection indexed as Mc*(010) appears: this plane forms a 52° angle with Mc(01), has $d = 57$ nm, and consists of slightly sheared gibbsite columns (Fig. 2g). Therefore, the gibbsite mesocrystal exhibits a monoclinic structure with $a_{Mc} = 93$ nm, $b_{Mc} = \sqrt{3}a$, $c_{Mc} \approx$ (gibbsite thickness)/(sin $\alpha$), $\alpha = 38°$, and $\beta = \gamma = 90°$.

Collectively, WAXS/SAXS, SEM, and TEM establish a uniform monoclinic framework in which nanoplates stack on Gb(001) and are displaced by 6–12 nm along Gb[010]. The resulting structure resembles the columnar phase observed in gibbsite platelets[15] and in disc-like molecules of liquid crystals[30]. However, in our system, the initial suspension of gibbsite is dilute (volume fraction = 0.81 vol.%) compared to the high-volume ratio (>45%) typically required for liquid-crystal phases[15]. LCTEM observation of the freshly prepared suspension (i.e., without aging) reveals the presence of staggered gibbsite stacks (Suppl. Fig. 4), indicating that mesocrystal formation arises from the self-assembly and sedimentation of nanoplates in this dilute system.

**The OA of gibbsite nanoparticles by sliding**
In situ LCTEM was employed to reveal the dynamic OA process between gibbsite nanoplates during the early stages of crystallization, leading to the formation of mesocrystals. A well-dispersed nanoplate suspension was sealed by two 30–50 nm thick electron-beam-transparent $Si_xN_y$ membranes, which enable the observation of liquids under high-vacuum conditions in TEM. Representative OA dynamics of gibbsite nanoplates in deionized water are presented in Supplementary Movies 1 and 2, Fig. 3, and Supplementary Fig. 6. The points of zero charge of bare $Si_xN_y$ and edge faces of gibbsite nanoplates are ~pH 4.1[31] and pH 9.0–10.0[26], respectively. Some gibbsite nanoplates initially adhered to the negatively charged $Si_xN_y$ membrane via the positively charged edge faces, driven by electrostatic attractions and van der Waals forces[32,33] (Fig. 3a). The electron beam was initially blanked to avoid beam-induced artifacts.

The in situ observations reveal a two-stage OA process: (i) rotation and jump-to-contact to achieve partial basal–basal overlap, and (ii) sliding primarily along the Gb[010] direction to maximize overlap toward a staggered configuration. After several seconds of electron-beam irradiation, edge-attached gibbsite nanoplate detached from the membrane, rotated, and flipped to a position of partial overlap with the closest gibbsite nanoplate via basal–basal attachment within ≈1 s. The initial detachment is likely due to the accumulation of positive charges on the $Si_xN_y$ surface from secondary electron emission[34]. The following motion prioritizes the alignment of nanoplates on their basal surfaces over maximizing the overlapping area, indicating the presence of strong, long-range attractive forces between their basal surfaces. These attractions potentially arise from a combination of hydration forces, ion-ion correlations, and van der Waals interactions[35].

After the flip at ≈11 s, the nanoplate—initially well aligned with the underlying one—underwent an in-plane rotation of ≈10° (Fig. 3b), likely caused by a local perturbation during the onset of sliding (e.g., minor surface asperities as shown in Suppl. Fig. 4). No collisions with neighboring particles were observed during this period. The misalignment was gradually corrected, consistent with PMF predictions[29] of a strong preference for basal–basal alignment within ≈10° and with SAED confirming the final alignment (Suppl. Fig. 2).

During stage (ii), the nanoplate slid mainly along the Gb[010] direction (Fig. 3b), increasing overlap while progressively slowing down (Fig. 3c). This sliding direction matches the tilting direction of nanoplate columns in mesocrystals, indicating a preferred pathway for alignment. Similar two-stage behavior was consistently observed

(Supplementary Movie 2), although sliding rates varied between events. These differences likely arise from variations in hydrodynamic drag due to local liquid confinement. Changes in the water-layer thickness between plates can strongly influence viscous resistance and particle mobility, as reported for nanoparticle diffusion in thin water films at solid−liquid interfaces[34,36]. The final offsets recorded in situ (18−30 nm, Fig. 3 and Suppl. Fig. 6) exceed the ensemble mean (≈12 nm) but fall within the upper tail of the ex situ distribution (Suppl. Fig. 10), suggesting contributions from statistical variability and restricted Brownian mobility in the liquid-cell gap.

We also observed that stacked gibbsite nanoplates are separated by several water layers when their basal planes are parallel to the beam (Suppl. Figs. 4 and 5, and Supplementary Movie 3), consistent with simulations showing a second local minimum in the free-energy profile corresponding to a solvent-separated state[29,37]. Moreover, the movie shows that the platelets are not perfectly flat, leading to variations in the separation. This solvent-separated state and limited area of closest approach enable the feasibility of the sliding motion. In stage 2, the overlying particle moves in the Gb[010] direction relative to the contacting particle to increase overlapping surface area, probably driven by van der Waals attraction and the energy gain associated with increasing the robustness of the interparticle hydrogen bonding network. This implies an energy landscape for sliding motion with barriers small enough to enable random walk configuration sampling by thermal motion, biased by the collective forces that favor overlap. We present below the results from molecular dynamics (MD) simulations for two solvent-separated gibbsite nanoplates in water aimed at providing a qualitative discussion of the following observations: (i) the mutual sliding motion is favored along the Gb[010] direction; (ii) the energy barriers for sliding motion increase with increasing overlap; and (iii) the lowest energy overlap entails an offset of the nanoplates rather than complete alignment, as shown in the mesocrystals.

## The energy landscape of the sliding

MD simulations were conducted to investigate the energy-structure relationship governing the sliding motion of two coplanar gibbsite nanoplatelets in water along the [010] direction, compared with prior results for the [100] direction[29] (Fig. 4 and Suppl. Fig. 7). The potential of mean force (PMF) was calculated using the umbrella sampling technique[38]. During sliding along the [010] direction, the red particle moves in the $y$ direction, increasing its overlap with the stationary blue particle (Fig. 4a), while maintaining a water-layer separation in the $z$ direction, consistent with our previous work[37].

The PMF profiles in Fig. 4 indicate that gibbsite particles in a solvent-separated state preferentially slide along the [010] direction to achieve alignment.

First, when sliding along the [010] direction, the system shows periodic energy minima (A-G on Fig. 4b) spaced 5.11 Å apart, corresponding to the $b$ unit cell length of gibbsite. These minima correspond to the configurations demonstrated by the snapshots A-G in Fig. 4d. These configurations illustrate the perfect alignment of particles with increasing overlapping surface area, from 1 hexagonal unit at A (see Suppl. Fig. 7, right panel for the definition of a hexagonal unit) to 37 at G, in the sequence of 1 < 4 < 9 < 16 < 23 < 30 < 37. Similarly, for sliding along the [100] direction, the PMF profile reveals minima I−IV (≈8.74 Å apart, matching the $a$ unit cell length), with snapshots I−IV (Fig. 4c) showing increasing alignment and overlapping hexagonal units in the order of 4 < 13 < 24 < 37.

Second, when sliding from one aligned configuration to another, the system must overcome energy barriers. For instance, advancing from C to D requires surpassing two barriers, with the highest being ≈4.44 kcal mol⁻¹ (the free-energy difference between C and C′). Similarly, progressing from II to III involves multiple barriers, with the highest at ≈10.50 kcal mol⁻¹ (between II′ and II″). Table 1 summarizes the highest energy barriers for transitions between aligned

configurations with increasing overlap of hexagonal units, while Fig. 5 plots the energy barriers per hexagonal unit.

Results indicate that achieving the first perfectly aligned configuration (A or I) requires a smaller energy barrier when sliding along the [010] direction compared to the [100] direction (0.89 vs. 2.49 kcal mol⁻¹). In configuration A, the two particles share one overlapping hexagonal unit, while in configuration I, they share four. This smaller initial barrier likely promotes alignment along the [010] direction (configuration A) over the [100] direction (configuration I).

Additionally, transitioning to the fully aligned configuration (G or IV) involves seven steps (A to G) in the [010] direction and four steps (I to IV) in the [100] direction. In the 7-step process, the number of new hexagonal units added per step is 1, 3, 5, 7, 7, 7, and 7, with energy barriers of 0.89, 1.82, 2.56, 4.44, 5.25, 5.83, and 6.15 kcal mol⁻¹. In the 4-step process, these numbers are 4, 9, 11, and 13, with barriers of 2.49, 5.42, 10.5, and 15.1 kcal mol⁻¹. Dividing the motion into more steps with fewer added hexagonal units lowers the energy barrier. Figure 5 further demonstrates that the energy barrier per overlapping hexagonal unit is lower in the [010] direction than in the [100] direction and decreases as the overlapped surface area increases. This supports the preference for sliding along the [010] direction over the [100] direction.

Third, when a particle slides to overlap with another, the energy barrier arises from two main contributions: the formation of a new overlapping region and sliding across the pre-existing overlapping region. In recent simulations (see Suppl. Fig. 8), we examined a gibbsite particle sliding entirely overlapped with a larger gibbsite surface along the [010] direction, where the barrier is solely due to sliding without any increase in the overlapped surface[39]. The PMF results (Suppl. Fig. 8) show an energy barrier of ≈5 kcal mol⁻¹ for sliding from a misaligned to an aligned configuration. Normalizing this barrier by the particle's 37 hexagonal units yields the black line in Fig. 5. Comparison of the blue and red lines in Fig. 5 shows that during the early stages of sliding, when new surface area is added (e.g., to A or from A to B), this contribution significantly increases the energy barrier, as reflected by the higher blue line. At later stages, when the previously overlapped surface dominates, the blue line approaches the black line. This suggests that the barrier per unit area for creating a new overlapped surface is higher than that for sliding along the existing overlapped interface. Consequently, the increase in energy barriers predicts slower sliding kinetics as the overlapping surface area grows, consistent with the experimental results in Fig. 3c. Note that in our simulation, the particle size (≈4.2 nm in diameter) is significantly smaller than that in the experiments (≈90 nm). This precludes quantitative energy barrier predictions for particle sliding in the experiments, as the relationship between the energy barrier and particle size is unknown. Additionally, the simulations assume a uniform water thickness between perfectly flat platelets, whereas TEM data (Suppl. Fig. 4) show that real platelets are not perfectly flat across their entire diameter. Although the high-resolution images in the liquid-cell (Suppl. Fig. 5) indicate an inter-basal separation of ≈0.7 nm, consistent with two molecular layers of water, the contrast and projection effects inherent to LCTEM prevent precise point-by-point measurement of the water-layer thickness. To better match the experiments, we perform another simulation where two particles are separated by two water layers and have positively charged edges (2 W). The PMF profile (Suppl. Fig. 9) indicates a reduction in energy barriers for 2 W compared to 1 W. This is expected, as the barriers in the PMFs arise from the hydration structure at the interface. Such barriers are known to decay rapidly as the separation increases to multiple water layers, typically becoming negligible at a separation of just three or four layers, or ≈ 1 nm[11]. Due to these discrepancies and uncertainties, the simulation results are primarily used for qualitative prediction of the free-energy landscape of the particle sliding.

Finally, the PMF profiles for both neutral particles (Fig. 4) and positively charged edges (Suppl. Fig. 9) provide insights into the

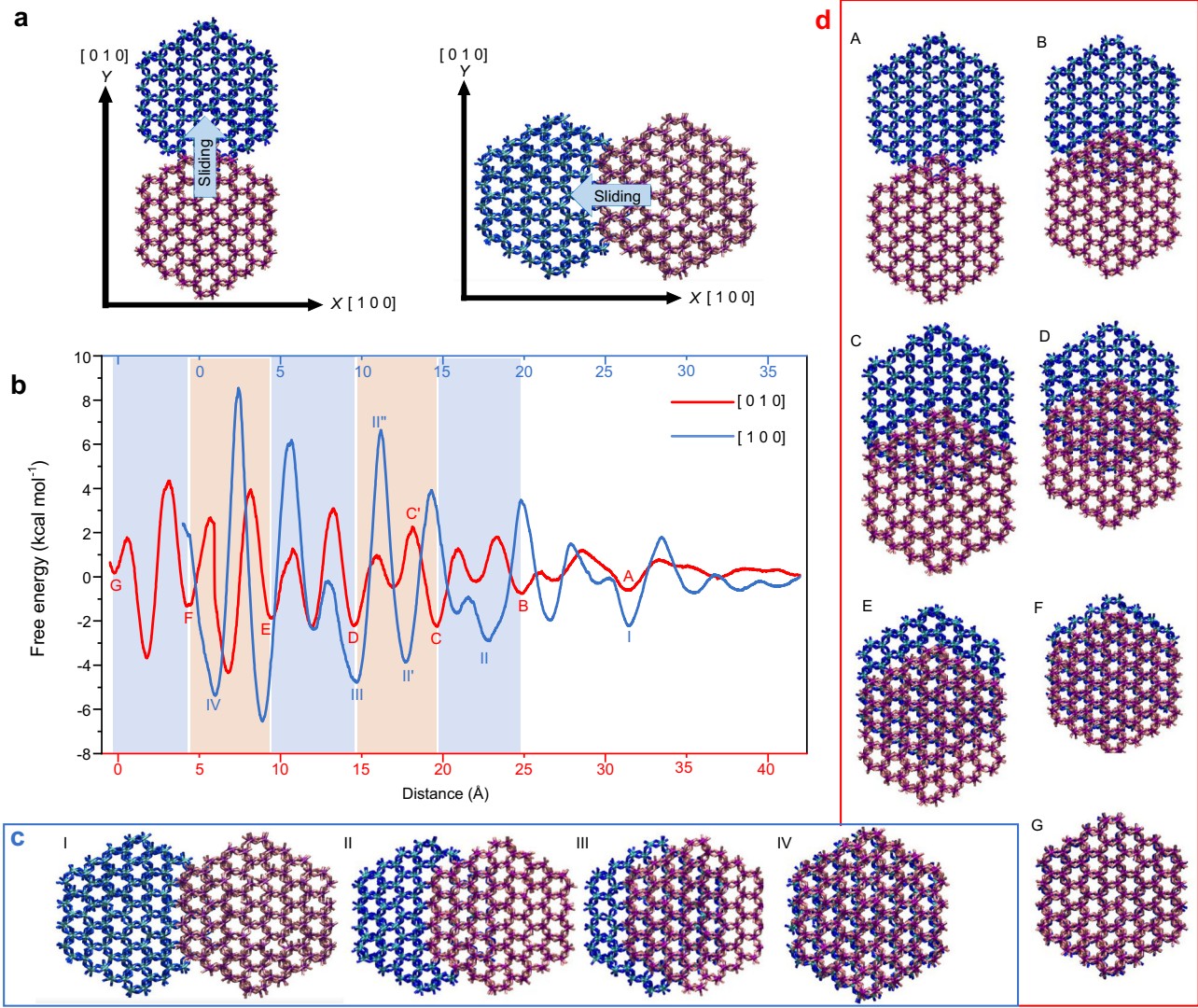

**Fig. 4 | Potential of mean force for the sliding motion of two coplanar gibbsite nanoplatelets. a** Sliding motions of the red gibbsite nanoparticles in water along the [010] and [100] directions toward alignment and overlap with the blue particle. The two particles remain separated in the z direction by a water layer throughout the PMF calculations. **b** Potential of mean force as a function of center of mass distance between two particles during sliding motions along the [010] and [100]

directions (i.e., overlap increasing to the left). Shade areas demonstrate the periodicity of the red PMF profile. **c** Simulation snapshots I-IV demonstrate the particle configurations at points I–IV marked on the blue PMF profile. **d** Simulation snapshots A-G illustrate the particle configurations (water is not shown) at points A–G marked on the red PMF profile. Source data are provided as a Source Data file.

**Table 1 | Energy barrier required to advance from one alignment to another**

| [010] direction | to A | A to B | B to C | C to D | D to E | E to F | F to G |
|---|---|---|---|---|---|---|---|
| Number of overlapped hexagonal units at the end | 1 | 4 | 9 | 16 | 23 | 30 | 37 |
| Number of new overlapped hexagons added | 1 | 3 ( = 4–1) | 5 ( = 9–4) | 7 | 7 | 7 | 7 |
| Highest energy barrier (kcal mol⁻¹) | 0.89 | 1.82 | 2.56 | 4.44 | 5.25 | 5.83 | 6.15 |
| Barrier per overlapped hexagon | 0.89 | 0.45( = 1.82/4) | 0.28 | 0.28 | 0.23 | 0.19 | 0.17 |
| **[100] direction** | **to I** | **I–II** | **II–III** | **III–IV** | | | |
| Number of overlapped hexagons at the end | 4 | 13 | 24 | 37 | | | |
| Number of new overlapped hexagons added | 4 | 9 | 11 | 13 | | | |
| Highest energy barrier (kcal mol⁻¹) | 2.49 | 5.42 | 10.5 | 15.1 | | | |
| Barrier per overlapped hexagon | 0.62 | 0.42 | 0.44 | 0.41 | | | |

thermodynamic origin of the uniform stagger. When sliding along the [010] direction, the completely overlapped and aligned configuration (e.g., configuration G) is less energetically favorable relative to a staggered configuration, regardless of water thickness between the

two particles. This may result from a decrease in entropy due to an increase in the number of confined water molecules and the merging of two particles into one. Additionally, for the charge-neutral gibbsite particle, the discontinuity at the edge and the under-coordinated

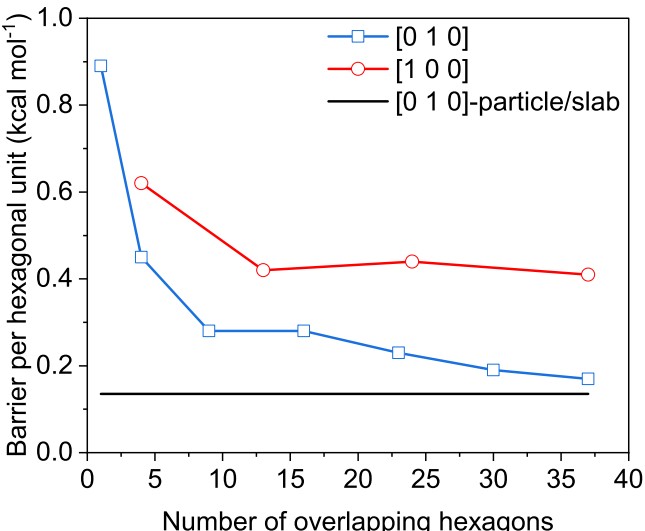

**Fig. 5 | Free energy barrier per overlapped hexagonal unit required for particles to transition to the next alignment configuration along the [010] (blue) and [100] (red) directions.** The black line shows the barrier when a particle slides on a surface with a constant overlapped area along the [010] direction (see Suppl. Fig. 8 for details). Source data are provided as a Source Data file.

edge-Al atoms (i.e., 5-fold coordinated Al; see Methods) introduce local charges. The staggered configuration likely mitigates electrostatic repulsion between like-charged edges. For the particles with positively charged edges, the electrostatic repulsion is stronger than that of neutral particles. This reinforces the results that the staggered configuration should dominate. The extent of particle staggering will depend on the balance among electrostatic edge-edge repulsion, van der Waals attraction (both of which depend on particle size and distance between them), and the entropy associated with sliding. In our simulations, the global energy minimum along the [010] sliding path occurs at an offset of ≈0.7 nm from perfect overlap for small model particles (4.2 nm). When scaled to full-sized gibbsite nanoplates with a diagonal dimension of ≈93 nm, this corresponds to a projected offset of ≈15.5 nm, consistent with the offset distribution measured in SEM and TEM images (Suppl. Fig. 10), which yields an average of 11.5 ± 6 nm.

Gravity, which is neglected in our simulations, can shift this balance for sufficiently large platelets. Van der Heijden et al. demonstrated that gravity compacts columns formed by large gibbsite platelets (≈570 nm in diameter), resulting in denser packing, pronounced column undulations, and a misalignment between the platelet orientation and the column axis near the bottom of the structure[40]. In contrast, our single-aggregate SAXS patterns for 90 nm platelets (Fig. 2e) reveal a uniform, coherent tilt of the hexagonal columns with no orientational fluctuations. This consistency suggests that at the sub-100 nm length scale, van der Waals, electrostatic, and hydration forces dominate over gravitational torque, preserving columnar alignment.

Contrasting the behavior of Al(OH)$_3$ nanoplatelets presented here with that reported for rhombohedral nanoplatelets of boehmite (γ-AlOOH) with similar dimensions[41] further highlights the importance of the solvent-separated state in achieving long-range order. Despite the fact that, like the gibbsite particles of this study, the boehmite platelets possessed neutral faces and charged edges over the range of pH values investigated[14] and assembled by OA, the aggregates that formed were disordered, exhibiting a fractal dimension predicted for platelets that bind with a low area-fraction of overlap via diffusion-limited aggregation. However, cryoTEM analysis showed that the boehmite platelets within the aggregates exhibited solid-solid contacts with a coherent lattice across the particle-particle boundaries and no intervening

solvent[41]. The contrast between these two systems further highlights the role of hydration barriers in enabling gibbsite particle sliding and the achievement of long-range order.

By combining structural analysis, in situ observation, and molecular simulations, we demonstrate the importance of sliding motion in enabling mesocrystal formation by OA, using the gibbsite nanoplate system as a case study. The results reveal that the gibbsite nanoplate prefers to slide along the [010] direction on the basal surface of another nanoplate to increase the overlapping surface. The balance among electrostatic edge-edge repulsion, van der Waals attraction (both of which depend on particle size and distance between them), hydration force, and the entropy associated with sliding leads to the formation of the staggered gibbsite mesocrystals. Our findings link the mesoscale structure of gibbsite aggregates to the atomic-scale energy landscape of gibbsite nanoplate interactions, thereby expanding the understanding of forces and conditions that yield OA in two-dimensional material systems. Our results provide fundamental insights that could ultimately aid in the development of predictive models for OA and the stability of various materials.

## Methods

### Synthesis and characterization of gibbsite nanoplates
Gibbsite nanoplates were synthesized using a two-step method[24]. A 0.25 M aluminum (Al) solution was prepared by dissolving aluminum nitrate nonahydrate [Al(NO$_3$)$_3$·9H$_2$O, ≥98%, Sigma-Aldrich] in Milli-Q water (pH 5.6, 18.20 MΩ cm$^{-1}$) while stirring. The pH was adjusted to 5.0 using a 3 M sodium hydroxide (NaOH) aqueous solution (98%, Sigma-Aldrich). The mixture was continuously stirred for 1 h, after which the gel-like precipitates were collected by centrifugation. These precipitates were washed three times with deionized water and then dispersed in deionized water to form 0.25 M Al suspensions. Subsequently, 16 mL of suspension was transferred to a 20 mL Teflon vessel, sealed inside a Parr bomb, and heated at 80 °C for 5 days. The resultant white product was collected by centrifugation (10,000 × g, 30 min), washed three times with Milli-Q water, and resuspended in Milli-Q water.

To prevent irreversible aggregation of particles during the drying process, gibbsite nanoparticles were kept in an aqueous solution without undergoing any drying before mesocrystal formation. Initially, gibbsite nanoplates were well dispersed in Milli-Q water after 30 min of intense ultrasonication, resulting in a dilute suspension (volume fraction = 0.81 vol.%). A 40 mL suspension of gibbsite nanoparticles was left undisturbed at 20 °C for 6 months in a 50 mL centrifuge tube. This resulted in the formation of mesocrystals at the bottom. To collect the formed mesocrystals, the sediment at the bottom of the centrifuge tube was transferred into DI water using a pipette. After allowing the particles to settle, this process was repeated multiple times, resulting in the separation of mesocrystals ranging in size from tens to hundreds of micrometers.

### Wide- and small-angle x-ray scattering
The single-crystal XRD experiments were conducted on a Bruker D8 QUEST diffractometer, equipped with an Incoatec IµS 3.0 Microfocus Mo X-ray source, a PHOTON II detector, and a KAPPA goniometer. WAXS/SAXS experiments of the single mesocrystal were conducted at the Complex Materials Scattering (CMS, 11-BM) beamline at the National Synchrotron Light Source II (NSLS-II) at Brookhaven National Laboratory (BNL). Samples were prepared without ultrasonication, ruling out beam-induced structural artifacts. The sample was loaded onto the MiTeGen MicroMeshes tip and mounted on a 5-degree-of-freedom goniometer to enable rotation of the sample to different crystallographic orientations. An X-ray beam with photon energy of E = 13.5 keV was used for scattering measurements. A Pilatus 2 M detector collected SAXS patterns with a sample-to-detector distance of 5.04 m. Simultaneously, WAXS was recorded with a Pilatus 800 K with a sample-to-detector distance of 0.26 m.

## Electron microscopy

SEM characterization was carried out in FIB-SEM (Helios NanoLab 600i, FEI, Hillsboro, OR). A carbon coater was used to deposit carbon thin films of 2–3 nm in thickness to enhance imaging. TEM characterizations were performed on a 300 kV FEI Titan environmental TEM equipped with a Gatan Metro 300 camera. Liquid-cell TEM was conducted using a commercial liquid holder and liquid cells from Hummingbird Scientific, in which two square silicon chips, each hosting 50 nm thick silicon nitride membranes, were separated by 500 nm gold spacers deposited around the edges. Before loading the solution, the membranes underwent a one-minute $Ar_2/O_2$ plasma cleaning to eliminate organic contaminants and make the surfaces hydrophilic. The aqueous suspension of gibbsite, with a concentration of $1\,mg\,L^{-1}$ and a pH of 5.6, was prepared by sonicating dried powder in deionized water for 30 min. A 0.5 µL volume of this solution was dropped between the two membranes, then loaded into the static liquid cell for vacuum testing in a Pfeiffer vacuum chamber. To minimize the bowing effect of the $SiN_x$ windows, we primarily recorded movies near the edges, where the liquid layer was thinnest. The particle contours in each frame were extracted using ImageJ[42]. The translational displacement centroid positions were calculated relative to the initial frame according to detailed contours. Instantaneous speed was obtained via a 3-point central difference method, which minimizes random frame-to-frame noise. Rotations were quantified using the Kabsch algorithm, which computes the optimal rigid-body rotation between successive contours, with uncertainties derived from bootstrapping the contour coordinates.

## Molecular simulation

The method used here is similar to that used in our previous paper[29]. We calculate the PMF using the COLVARS[43] package available in LAMMPS[44]. Before performing a PMF calculation, we equilibrate the systems containing two particles and water molecules (inserted on a grid) in an *NPT* (constant number of atoms, pressure 1 atm, and temperature 300 K) ensemble (see Suppl. Fig. 7, left panel for the equilibrated system). Next, we perform steered MD (SMD) simulations to obtain the configurations of the system along the reaction coordinates for the PMF calculation. In the steered MD simulations, the center of mass (COM) of one particle is kept fixed by excluding eight atoms from the integration of the equation of motion[45]. The second particle translates with a constant velocity of $10\,Å\,ns^{-1}$ in the sliding motions with a force constant of $5000\,kcal\,mol^{-1}\,Å^{-2}$. Note that the PMF results should not depend on the parameters chosen for SMD simulations, i.e., the SMD simulation is used to obtain the configurations for subsequent PMF calculations. The umbrella sampling method (US) is applied to calculate PMF profiles with a separation between windows of 0.025 Å and a force constant of $5000\,kcal\,mol^{-1}\,Å^{-2}$) for sliding motions. At each window, the simulation is conducted for 3.0 ns. The large force constant is needed to keep the two particles fluctuating around a specific reaction coordinate, and the small window separation is required to obtain sufficient overlap among windows for the convergence of the PMF calculation. The weighted histogram analysis method is applied to extract the PMF profile from the simulation trajectory[46]. During the US simulations, the center of mass of the red particle is not allowed to move in the *x* and *z* directions by zeroing its momentum every 100 time steps using the 'fix' momentum in LAMMPS[44].

The pseudohexagonal gibbsite particle (Suppl. Fig. 7, top right panel) has 21.0 Å edges and a thickness of 13.4 Å (3 gibbsite layers). The basal surface is much smaller than that of natural or synthesized gibbsite crystals[47–50]. However, our particle size was selected as a reasonable choice to perform the expensive PMF calculation using the US method. This particle was carved out of a gibbsite slab along the (1 0 0) and (1 1 0) directions, see refs. 51,52. The resultant particle has an edge characterized by fivefold coordinated Al atoms. When the particle is placed in water, these Al atoms become 6-coordinated with a water

molecule. The -OH groups on all crystal faces and corners of our gibbsite particles allow for the formation of hydrogen bonds (H-bonds) between the two particles. Note that protonation/deprotonation can occur on aluminum (hydr)oxide surfaces (e.g., $\gamma$-$Al_2O_3$[53], gibbsite[28]). However, in this work, no protonation/deprotonation is allowed. The particles in Fig. 4 have both basal and edge surfaces that are neutral. In contrast, the particles in Suppl. Fig. 9 have positively charged edge surfaces with a charge density of $+0.1897\,C\,m^{-2}$. The charge is evenly distributed across the hydrogen atoms of the edge OH groups. To maintain overall charge neutrality in the simulation, $Cl^-$ ions are introduced to balance the positive surface charge. There are 4032 atoms belonging to gibbsite particles and 18,656 water molecules in the simulation box. In our simulations, all atoms can move freely, which means the particles can deform. However, because our gibbsite particle is small, particle bending is not as obvious as in the molecular ribbon model simulation[54], peeling simulation[55], or experiments for montmorillonite[56]. The center of mass of the particle is not allowed to move in the *x* and *z* directions during the PMF calculation.

The gibbsite particle is simulated using the ClayFF force field[57] with additional Al-O-H angle terms describing edge OH groups[58]. Water molecules are simulated using a flexible SPC water model[59]. The pairwise Lennard-Jones (LJ) potential energy is expressed as: $V_{LJ} = 4\varepsilon[(\frac{\sigma}{r})^{12} - (\frac{\sigma}{r})^{6}]$, where *r* is the distance between two atoms, $\varepsilon$ and $\sigma$ are the depth of the potential energy well and the distance at which the LJ potential is zero, respectively. LJ interactions between different atom types are calculated using the Lorentz-Berthelot mixing rules $\varepsilon_{ij} = \sqrt{\varepsilon_{ii}\varepsilon_{jj}}$ and $\sigma_{ij} = (\sigma_{ii} + \sigma_{jj})/2$. Short-range interactions are calculated using a cutoff distance of 10. Long-range electrostatic interactions are computed using the PPPM (particle-particle-particle-mesh) solver[60]. Periodic boundary conditions are applied in all directions. Simulations are performed at 300 K and 1 atm using the Nose-Hoover thermostat and barostat[61,62] with a timestep of 1.0 fs.

## Data availability

The data that support the findings of this study are available from the corresponding authors upon request. Source data are provided with this paper.

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

## Acknowledgements
This material is based on work supported by the Ion Dynamics in Radioactive Environments and Materials (IDREAM) program, an Energy Frontier Research Center funded by the U.S. Department of Energy, Office of Science, Basic Energy Sciences (FWP 68932). X.L., C.P., D.J.J., K.M.R., and X.Z. also acknowledge support from the U.S. Department of Energy (DOE), Office of Science, Basic Energy Sciences (BES), Chemical Sciences, Geosciences, and Biosciences Division through its Geosciences Program at Pacific Northwest National Laboratory (PNNL) (FWP 56674). PNNL is a multiprogram national laboratory operated by Battelle Memorial Institute under contract no. DE-AC05-76RL01830 for the DOE. T.A.H. acknowledges support by the DOE Office of Science, BES, Chemical Sciences, Geosciences, and Biosciences Division through its Geosciences Program at Sandia National Laboratories (SNL) (FWP 24-015452). This article was authored by an employee of National Technology & Engineering Solutions of Sandia, LLC, under contract no. DE-NA0003525 with the US DOE. This research also utilized the Complex Materials Scattering (CMS, 11-BM) beamline of the National Synchrotron Light Source II, a U.S. Department of Energy (DOE) Office of Science User Facility operated by the DOE Office of Science at Brookhaven National Laboratory under Contract No. DE-SC0012704. A portion of the work was carried out in the Environmental and Molecular Sciences Laboratory (EMSL), a national scientific user facility at PNNL sponsored by the DOE Office of Biological and Environmental Research, under user proposals 10.46936/lser.proj.2020.51382/60000186 and 10.46936/lser.-proj.2021.51922/60000373.

## Author contributions
X.L., K.M.R., and X.Z. designed the research. T.H. performed the MD simulations and analysis. H.Z. and R.L. carried out the S/WAXS measurements, and together with S.T.M. and X.L., analyzed the data. X.L. and L.L. conducted the LCTEM characterization and analysis, and X.L. and P.C. synthesized the mesocrystal. M.E.B. performed the XRD measurements. X.L., X.Z., and L.L. conducted the TEM and SEM characterizations. H.F., J.J.D.Y., and C.I.P. contributed to discussions and manuscript revision. The manuscript was written by X.L., T.H., K.M.R., and X.Z., with input from all authors.

## Competing interests
The authors declare no competing interests.
