## [Transparent Peer Review file · Nature Communications]

Mesocrystal growth through oriented sliding and attachment of nanoplates

Corresponding Author: Dr Xin Zhang

Version 0:

Reviewer comments:

Reviewer #1

(Remarks to the Author)

The formation of “mesocrystals”, also referred to as “supercrystals” or “supracrystals” in the literature, by the spontaneous ordered assembly of crystalline nanoparticles has recently attracted much interest worldwide, due to the appearance of new collective physical properties, for example. However, the formation mechanisms of mesocrystals are not yet well understood, which hinders their applications. One of these growth mechanisms, called “oriented attachment”, consists in the rotation and translation of each incoming nanometric crystalline particle to fit onto the growing mesocrystal. The aim of this manuscript is to describe this mechanism in detail in the case of gibbsite platelets, using x-ray scattering at wide and small scattering angles, electron microscopy, and numerical simulations. In my opinion, its most outstanding feature is the real-time observation by liquid-cell TEM of the rotation/translation of a gibbsite platelet as it attaches to a mesocrystal. Similar observations have already been published in the case of compact semiconductor nanoparticles (e.g., CdSe), but, to the best of my knowledge, not yet in the case of plate-like insulating oxide nanocrystals (such as gibbsite). Therefore, this topic should be of interest to the broad readership of Nat. Comm.

I think that the most original part of this work is the in-situ liquid-cell TEM study, which would warrant publication in Nat. Comm. However, the evidence presented in this work consists of only one sequence of a few images, which casts doubt on the generality and reproducibility of the phenomenon. Although the conclusions drawn by the authors (staggered platelet stacking) are well supported by the x-ray scattering study of the final mesocrystals and the numerical simulations, the most exciting part of the work (the in-situ LC-TEM study) is not yet convincing, due to the rather limited data. For publication, I feel that more oriented attachment sequences need to be recorded, which would provide better statistics for comparison with the other techniques and more insight into this mechanism.

In addition, the x-ray scattering part, which is much more classical and straightforward, is a bit lengthy and should be shortened. Moreover, the numerical simulations were done with only one water layer between the platelets, although the data show that there are more. This could affect the H-bond network and modify the conclusions of the simulations, as already discussed by the authors. I suggest performing the same simulations with two and three water layers to check the robustness of the conclusions with respect to this parameter.

I also list some minor comments below:

- L 47 (and elsewhere): please, replace “rotational forces” by “torques”.
- L 113: replace “crossed polarized light microscopy” by “polarized-light optical microscopy” (POM).
- L 118: 10 – 20° mosaicity.
- Fig. 1f: the -3-1-1 label should be replaced by 3-1-1.
- L 164 (and elsewhere): replace “angel” by “angle”.
- L 166: replace (c,d) by (d,e).
- L 220: The comparison with the SmC phase is not very good because this phase is a 1-dimensional stack of liquid layers. In fact, disk-like liquid crystalline molecules sometimes form columnar phases with the stacking axis at any angle with respect to the 2-dimensional lattice of columns. (See for example, A.M.Levelut et al, J. Physique, vol 42, p 147 (1981)). By the way, it would be interesting to determine whether or not the mesocrystals show any hkl reflection with all indices different from zero. If so, these objects are truly crystalline. If not, this means that the columnar stacks are uncorrelated and the mesocrystals can be compared to columnar liquid crystals. (See also the papers by Petukhov and Lekkerkerker on the hexagonal LC phase of gibbsite platelets)
- L 265: Electrostatic interactions can be attractive, for example, in the case of ion correlations, as mentioned at the end of the sentence. Please, rephrase.
- Finally, the English style should be carefully edited in many places.

Reviewer #2

(Remarks to the Author)

Li et al. report a systematic study on the formation of mesocrystals of gibbsite by ordered attachment using a combination of in situ and ex-situ real-space imaging and scattering techniques as well as numerical simulations. The experiments provide a fantastically detailed characterization of the structure of the mesocrystals and make a very convincing case for this non-conventional crystal growth mechanism. They demonstrate in a – to my knowledge – unique manner the translational and orientational alignment of nano-platelets to form mesoscopic crystals. Overall, I find the experimental work very interesting but have some concerns regarding the relevance of the numerical simulations that should be clarified.

Main concern:

I understand that the platelets in the simulations have to be chosen smaller than in the experiments for computational reasons. While I am not so much worried about the reduced lateral dimensions, the decrease of the plate separation for 3-4nm in the experiments to 0.3nm seems rather critical. On p. 18, the authors rightfully state that 'the barriers to sliding ... in the experiments will be greatly reduced ...' compared to the simulations. Given the correlation length of water and the H-bonding network of about 1nm, shouldn't one rather expect that any anisotropy is completely washed out at 3-4nm distance? If so, what is then the relevance of the simulations for the experimental observations? This must be addressed very clearly.

Minor comments:

1. P. 13, l 255: I understand that the x-ray data show an (ensemble) average staggering of 6-12nm. The LCTEM shows 30nm. I presume that this is based on a single observation – or is this also a statistically averaged quantity? If so, what is the error bar? If not, isn't then simply the lack of statistics an equally likely explanation as the proposed hindered diffusion due to the walls?
2. Do I understand correctly that the platelets are forced to move strictly along the (010) and (100) direction without allowing neither lateral nor vertical displacements of the center-of-mass nor orientational fluctuations? Isn't that very somewhat unrealistic, in particular if the distance between platelets is 3-4nm? I would expect a substantial reduction of the relevant barriers if such fluctuations would be included.
3. P. 19, line 392: 'The contrast ... highlights the role of hydration barriers ...': again, I don't see the evidence for this beyond the simulations, which are in a different regime than the experiments. (see main concern above)
4. P. 5, line 104: AFM measurements reported attractive forces between the gibbsite basal plane and (negatively charged) silica tips below pH 6 indicating positive zeta potentials and surface charges (see Gan&Franks, Langmuir 2006; Klaassen et al. Nanoscale 2017)
5. P. 11, line 207: does the 'direction rather than positional order' mean that the d-spacing is not well-defined? (I may misunderstand something here.)

Typos:

- p. 13. Line 269: 'basal planes'
p. 6, line 128: '... [100] and [100]...' ???
p. 7, caption Fig. 1: caption speaks of 'the mesocrystal', the text of 'a mesocrystal'. Do all images show the same or just 'a' mesocrystal?
p. 7: Fig. 1c: dark blue labels are invisible when printed on paper
p. 9: Fig. 2 cation : '...wide/small angle ...' or angle ;-)

Reviewer #3

(Remarks to the Author)

Reviewer #4

(Remarks to the Author)

The manuscript by Xiaoxu Li et al presents a remarkable experimental work on the formation and structure of a mesocrystal of charged gibbsite particles in deionized water. The work combined WAXS/SAXS/SEM/TEM/LCTEM to carefully elucidate the resulting crystalline ordered mesostructure. The manuscript is also very well written, which does not detract from the rest. Finally, the manuscript presents simulations and relies on them to try to explain the formation of the observed mesocrystal. It is on this last point that my problem is concentrated. The simulations are based on the observation (and not the other way around) that the particles slide over each other and attempt to show, in a way that I find unconvincing, why they stop sliding before perfect overlap, thus giving rise to the observed angle between the normal to the hexagonally ordered columns and the normal to the basal plane of the platelets. Guided by these observations, the authors chose to use molecular simulations to measure the free energy landscape of interaction between two particles of gibbsite forced to slide in relation to each other in two crystallographic directions while maintaining a constant basal distance. As might be expected, the free energy profile shows oscillations alternating between attraction and repulsion, which increase with the overlap of the particles. The authors

conclude that perfect overlap is not achieved for kinetic reasons, that is particles stop sliding because the free energy barrier increases with the particle overlap. But this explanation contradicts another remark made in the manuscript: "The uniformity of the lattice parameters suggests that the staggered arrangement of platelets represents a thermodynamic minimum in the degree of overlap". The authors have to choose: either the structure is kinetically arrested or it is thermodynamically stable, but it cannot be both at the same time. Beyond the semantics, the simulations carried out have a number of flaws: the positive electrostatic charges at the edge of the particles and the many-body interactions in the mesostructure are ignored. However, the repulsive electrostatic interactions coupled with the many-body interactions could very well explain the staggered arrangement of the particles, see for example the article by Delhomme et al (ref 16). As Helmut Coelfen pointed out, "A temporary stabilization of the nanoparticles is crucial for the nanoparticle orientation to a mesocrystal by interaction forces to maintain that the system is able to find the point of minimal energy on its energy landscape" (DOI: 10.1002/adma.200901365). Following the same logic, it would have been appropriate to try to determine the minimum free energy of interaction between the two particles. Incidentally, it can be noted that the minimum free energy of the two calculated profiles provided in the manuscript does not correspond to a perfect particle overlap (for the simulated system presented, this would result in an angle of about forty degrees if my calculations are correct). Finally, the authors forget the role of the gravitational field. Although it is negligible for the simulated system considered, it is one of the main drivers of the sedimentation of gibbsite particles and therefore of the formation of the observed mesocrystals. On the same subject, the authors may wish to read the work of Lekkerkerker and in particular these two articles (DOI: 10.1140/epjst/e2013-02075-x; <https://pubs.acs.org/doi/10.1021/la100797x>) observing the role of gravity on the formation of hexagonal columnar phase where average platelet orientation is decoupled from the column axis. To conclude I find the experimental study and its results very interesting, but the conclusions drawn from the simulations are not convincing. At the very least, the latter need to be moderated and discussed, particularly with regard to the numerous approximations (neglected physics) and hypotheses on which they are based.

Version 1:

Reviewer comments:

Reviewer #1

(Remarks to the Author)

The authors have revised their manuscript according to my comments. In particular, they have reproduced the in-situ TEM experiments and shortened the X-ray scattering section. They have also addressed my minor comments. I am not familiar enough with molecular simulations to properly assess this part, but I believe that the in-situ TEM study, on its own, is novel and exciting enough to justify publication of this article in Nature Communications. Therefore, I recommend publication of this revised manuscript with no further changes.

Reviewer #2

(Remarks to the Author)

I appreciate the efforts of the authors regarding the repeated experiments and the attempt to reconcile the experimental spacing of the basal planes and the one used in the simulations, as well as the charge along the edges. I watched the new (and the old) TEM movie over and over again in an attempt to convince myself whether it is fair to draw far-reaching conclusions based on these data. My honest opinion is not quite. I understand that recording these data is a great achievement as such – and I do also see that both videos clearly demonstrate a process in which an initially vertically standing platelet lies down flat and subsequently moves more or less in plane. However, whether that motion is along specific crystallographic axes or not, cannot be extracted from this very limited data set in convincing manner. The original movie simply contains three snapshots only, implying two consecutive steps in the same direction. The new one, upon sliding the movie back and forth from $t=11$ s to approx. 33s shows more a kind of rotation around the top corner than a directed motion along any specific crystallographic axis. In both cases, there are also adjacent particles to the bottom left (old movie) and on the bottom right (new movie) with which the moving particle seems to interact/collide. Based on the data presented, this might just as well be the cause for the change of direction as the suggested anisotropic hydration structure of the water in between the platelets. The case for the intrinsic directional diffusion would be more convincing if there were no neighbouring particles to interact with.

I understand that it is extremely difficult to carry out and interpret these experiments. Regarding my original main concern, the on-edge views of the particle stacks indeed show that spacings much thinner than the original estimate of 3-4nm do indeed occur with values that are compatible with two monolayers of H₂O. This is an important addition. I understand that it is impossible to extract the actual distances from the top view TEM images in Figs. 3 and S6.

While the authors state that the new data set in Fig. S6 is 'fully consistent with our original observation', the translation speed reported in Fig. S6h is 100x slower than in Fig. 3f for a very similar size of the particles. It does not become clear to me why this is 'fully consistent'. (Possibly, the separation for the old data is indeed 3-4nm as originally suggested whereas for the new ones it is indeed much thinner leading to stronger friction – but this is all speculative.)

Regarding Fig. 3f, I also don't understand the first data point at $t=2.4$ s. If I understand correctly, this data point originates from Δr_1 in Fig. 3b. However, that first step of the center of mass translation involves not only a translation but also the re-orientation process, which probably makes up for an important fraction of Δr_1 . It is therefore not justified to treat this data point on the same footage as the two subsequent ones. With only two data points left, however, the statement about the gradually decreasing lateral sliding speed with increasing particle becomes a bit futile.

I am also a bit confused about this new statement:

'Notably, in our simulations, the global energy minimum along the [010] sliding path occurs at an offset of approximately 0.7 nm from perfect overlap for small model particles (4.2 nm). When scaled to full-sized gibbsite nanoplates with a diagonal dimension of ~93 nm, this corresponds to a projected offset of ~15.5 nm, consistent with the offset distribution measured in SEM and TEM images (Fig. S10), which yields an average of 11.5 ± 6 nm.'

Assuming that the water layer thickness is indeed 0.7nm, the suggested scaling would imply that the staggering angle scales with the particle size. If, however, the staggering angle is caused by the electrostatic edge-to-edge repulsion, why would this scale with the particle size?

Taking everything together and looking at the raw data, I don't feel that the amount and detail of LC-TEM data presented – notwithstanding its beauty – is sufficient to support the conclusion of a strongly directional and hydration-mediated lateral translation mode.

Reviewer #3

(Remarks to the Author)

Reviewer #4

(Remarks to the Author)

The authors have responded satisfactorily to my comments. I have no further comments to make.

Version 2:

Reviewer comments:

Reviewer #2

(Remarks to the Author)

I think that the authors included now the necessary detail and due care in their arguments such that the readers can think for themselves about the great experimental results - and the remaining limitations that will have to be resolved in follow up research of this great work.

I recommend publication as is.

Point-by-Point Response

General improvement

The authors appreciate the valuable comments from all four reviewers. These comments are crucial for further improving the quality of our manuscript. Below, we provide detailed replies to each of the referees' comments (shown in *italic blue*) along with the changes made in our revised manuscript. The corresponding line numbers in the revised manuscript, in which we have tracked the changes, are also indicated. When substantial modifications were necessary, we included the new text below for ease of assessment. We hope that the revised version of our manuscript is now suitable for publication in *Nature Communications*.

Reviewer #1:

Comment 1:

The formation of “mesocrystals”, also referred to as “supercrystals” or “supracrystals” in the literature, by the spontaneous ordered assembly of crystalline nanoparticles has recently attracted much interest worldwide, due to the appearance of new collective physical properties, for example. However, the formation mechanisms of mesocrystals are not yet well understood, which hinders their applications. One of these growth mechanisms, called “oriented attachment”, consists in the rotation and translation of each incoming nanometric crystalline particle to fit onto the growing mesocrystal. The aim of this manuscript is to describe this mechanism in detail in the case of gibbsite platelets, using x-ray scattering at wide and small scattering angles, electron microscopy, and numerical simulations. In my opinion, its most outstanding feature is the real-time observation by liquid-cell TEM of the rotation/translation of a gibbsite platelet as it attaches to a mesocrystal. Similar observations have already been published in the case of compact semiconductor nanoparticles (e.g., CdSe), but, to the best of my knowledge, not yet in the case of plate-like insulating oxide nanocrystals (such as gibbsite). Therefore, this topic should be of interest to the broad readership of Nat. Comm.

Response:

We sincerely thank the reviewer for their thoughtful and encouraging comments. We are particularly grateful for their recognition of the significance of real-time liquid-cell TEM observation of oriented attachment in plate-like insulating oxides.

Comment 2:

I think that the most original part of this work is the in-situ liquid-cell TEM study, which would warrant publication in Nat. Comm. However, the evidence presented in this work consists of only one sequence of a few images, which casts doubt on the generality and reproducibility of the phenomenon. Although the conclusions drawn by the authors (staggered platelet stacking) are well supported by the x-ray scattering study of the final mesocrystals and the numerical simulations, the most exciting part of the work (the in-situ LC-TEM study) is not yet convincing, due to the rather limited data. For publication, I feel that more oriented attachment sequences need to be recorded, which would provide better statistics for comparison with the other techniques and more insight into this mechanism.

Response:

We agree that the limited dataset in the original submission could raise concerns regarding reproducibility and the generality of the observed phenomenon. In direct response to this comment, we conducted new LC-TEM experiments under identical experimental conditions. The newly acquired image sequences and a supplementary video have now been included as new Fig. S6 and Supplementary Movie S2, respectively.

As shown in Fig. S6a–f, the newly recorded time-lapse TEM images demonstrate two gibbsite nanoplates approaching and attaching in a manner fully consistent with our original observation. The trajectory of the center of mass following nanoplate flipping ($t = 11.0$ s) is plotted in Fig. S6g, which confirms a sliding motion first along the [010] axis and then toward the crystallographically equivalent $[-130]$ axis. Furthermore, Fig. S6h shows the averaged sliding speed versus time, revealing a progressive deceleration as the overlap area increases, further supporting our interpretation of a directional attachment process.

These new LC-TEM measurements not only confirm the reproducibility of the observed mechanism but also reinforce the robustness and generality of our conclusions. The consistency between the original and new observations strengthens the overall narrative by aligning well with the X-ray scattering analysis and numerical simulations presented in the manuscript.

Changes in Manuscript:

Figure S6 has been added in the SI.

Fig. S6. *In situ* liquid-cell TEM visualization of the oriented attachment (OA) of gibbsite nanoplates. (a–f) Time-lapse TEM images from Movie S2 of two nanoplates as they approach and attach. White dots mark the moving nanoplate’s center of mass projected onto the imaging plane, and white lines connect its current position of the center-of-mass to the earlier positions. (g) Detailed trajectory of the center of mass of the attaching nanoplate after it flips into partial alignment ($t = 11.0$ s). During sliding, the nanoplate changes its direction of motion from the [010] axis to the [1-30] axis. The [010] and [1-30] directions are crystallographically equivalent. (h) Average sliding speed of the attaching nanoplate versus time, showing that its velocity gradually decreases as time passes, and the overlap area increases.

Main text, lines 211-212: A new paragraph was added to summarize the results of the newly conducted experiments.

Previous text: “The typical motion of gibbsite nanoplate aggregation in DI water is shown in Supporting Information movies S1.”

Revised text: “Representative OA dynamics of gibbsite nanoplates in deionized water are shown in Supplementary Movies S1 and S2 and Fig. S6.”

A new recorded LCTEM movie replaces the SI movie S2.

Movie S2 *In situ* liquid cell TEM observation of the OA of gibbsite nanoplates (movie #2).

Comment 3:

In addition, the x-ray scattering part, which is much more classical and straightforward, is a bit lengthy and should be shortened.

Response:

We agree that the X-ray-scattering section can be streamlined. In response, we reduced it from ~500 to ~280 words by merging repetitive descriptions and relocating technical details to the Methods section, while retaining all essential experimental information and conclusions.

Changes in Manuscript:

Main text, Line 176-197:

Revised discussion:

“To resolve the long-range order of the gibbsite mesocrystals and its linkage to the crystallographic orientation of the building blocks, we analyzed a single aggregate with synchrotron wide- and small-angle X-ray scattering (WAXS/SAXS). We denote the gibbsite lattice indices as “Gb” and mesocrystal indices as “Mc.”

When the incident beam is aligned close to the Gb[001] axis (**Figs. 2a,b**), WAXS shows sharp spots on a faint Debye ring, and SAED confirms quasi-single-crystal order with an azimuthal mosaicity of $< 10^\circ$ (**Fig. S2**). This angular range matches the potential-of-mean-force minimum for in-plane rotation predicted by Ho et al., confirming that basal–basal interactions constrain the platelets within $\sim 10^\circ$ of perfect alignment²⁰. The concurrent SAXS pattern shows clear six-fold positional symmetry with a nearest-neighbor distance of 93 nm, corresponding to the edge-to-edge size of individual nanoplates (**Fig. 2c**). The pattern indexes to a centered rectangular 2D lattice with $a_{Mc} = 93$ nm and $b_{Mc} = \sqrt{3} a_{Mc}$ (**Fig. 2f**).

After rotating the aggregate by $\sim 90^\circ$ (**Figs. 2d,e**), the WAXS reflections are indexed to zone axes Gb[010]/[130] and Gb[100]/[110], confirming that the beam now lies parallel to the platelet basal planes. Diamond-shaped equatorial streaks in SAXS reveal anisotropic diffuse scattering along Gb[002], consistent with the stacking direction. The absence of $Mc^*(001)$ reflections indicates orientational alignment of gibbsite nanoplates, without clear long-range translational periodicity. In addition to the 2D $Mc(01)$ spots, a new reflection indexed as $Mc^*(010)$ appears: this plane forms a 52° angle with $Mc(01)$, has $d = 57$ nm, and consists of slightly sheared gibbsite columns (**Fig. 2g**). Therefore, the gibbsite mesocrystal exhibits a monoclinic structure with $a_{Mc} = 93$ nm, $b_{Mc} = \sqrt{3}a$, $c \approx (\text{gibbsite thickness})/\sin \alpha$, $\alpha = 38^\circ$, and $\beta = \gamma = 90^\circ$.”

Main text, lines 425-426, add details of sample preparation in methods:

Added sentences:

“Samples were prepared without ultrasonication, ruling out beam-induced structural artifacts.”

Comment 4:

Moreover, the numerical simulations were done with only one water layer between the platelets, although the data show that there are more. This could affect the H-bond network and modify the conclusions of the simulations, as already discussed by the authors. I suggest performing the same simulations with two and three water layers to check the robustness of the conclusions with respect to this parameter.

Response:

We appreciate the reviewer’s concern regarding the applicability of the PMF analysis to platelets separated by multiple water layers. To better estimate the actual separation, we performed high-resolution LCTEM imaging of gibbsite stacks in DI water (Fig. S5), which reveals inter-basal spacings of approximately 0.7 nm, consistent with two molecular layers of water. However, due to projection and contrast effects inherent in LCTEM, precise measurement of point-to-point water layer thickness remains challenging.

In direct response to this comment and those of other reviewers, we performed additional simulations and PMF calculations for two particles separated by two intervening water layers (2W), where both particles have positively charged edge surfaces. The purpose here is not to investigate the effect of water thickness or surface charge at the edges on the energy landscape, but rather to design a simulation that better matches the experimental data. The resulting PMF data are presented in Fig. S9, reproduced here for convenience.

These additional analyses confirm that our main conclusions remain valid, even for particles with positively charged edges and two water layers. Specifically:

1. The barrier during sliding motions increases, consistent with our original findings for neutral particles separated by a single water layer (1W).
2. The completely overlapped configuration does not correspond to the global minimum for both 1W and 2W systems.
3. Hydration effects remain crucial, as demonstrated by the reduced barrier for the 2W scenario compared to the 1W scenario.
4. The PMF for the 2W case exhibits fewer distinctive features than the 1W case, likely attributable to diminished water structuring effects between the particles.

We have incorporated these observations into the revised manuscript to highlight these points better and comprehensively address the reviewers' insights.

Changes in Manuscript:

Main text, lines 334-367:

“Note that in our simulation, the particle size (~ 4.2 nm in diameter) is much smaller than that in the experiments (~ 90 nm). This precludes quantitative energy barrier predictions for particle sliding in the experiments, as the relationship between the energy barrier and particle size is unknown. Additionally, the simulations assume a uniform water thickness between perfectly flat platelets, whereas TEM data (**Fig. S4**) show that real platelets are not perfectly flat across their entire diameter. Although the high-resolution images in the liquid cell (**Fig. S5**) indicate an inter-basal separation of roughly 0.7 nm—consistent with two molecular layers of water—the contrast and projection effects inherent to liquid-cell TEM prevent precise point-by-point measurement of the water-layer thickness. To better match the experiments, we perform another simulation where two particles are separated by two water layers and have positively charged edges (2W). The PMF profile (**Fig. S9**) indicates a reduction in energy barriers for 2W compared to 1W. This is expected, as the barriers in the PMFs arise from the hydration structure at the interface. Such barriers are known to decay rapidly as the separation increases to multiple water layers, typically becoming negligible at a separation of just three or four layers, or approximately 1 nm^{11} . Due to these discrepancies and uncertainties, the simulation results are primarily used for qualitative prediction of the free energy landscape of the particle sliding.

Finally, the PMF profiles for both neutral particles (**Fig. 4**) and positively charged edges (**Fig. S9**) provide insights into the thermodynamic origin of the uniform stagger. When sliding along the [010] direction, the completely overlapped and aligned configuration (e.g., configuration G) is less

energetically favorable relative to a staggered configuration, regardless of water thickness between the two particles. This may result from a decrease in entropy due to an increase in the number of confined water molecules and the merging of two particles into one. Additionally, for the charge-neutral gibbsite particle, the discontinuity at the edge and the under-coordinated edge-Al atoms (i.e., 5-fold coordinated Al; see Methods) introduce local charges. The staggered configuration likely mitigates electrostatic repulsion between like-charged edges. For the particles with positively charged edges, the electrostatic repulsion is stronger than that of neutral particles. This reinforces the results that the staggered configuration should dominate. The extent of particle staggering will depend on the balance among electrostatic edge-edge repulsion, van der Waals attraction (both of which depend on particle size and distance between them), and the entropy associated with sliding. Notably, in our simulations, the global energy minimum along the [010] sliding path occurs at an offset of approximately 0.7 nm from perfect overlap for small model particles (4.2 nm). When scaled to full-sized gibbsite nanoplates with a diagonal dimension of ~ 93 nm, this corresponds to a projected offset of ~ 15.5 nm, consistent with the offset distribution measured in SEM and TEM images (**Fig. S10**), which yields an average of 11.5 ± 6 nm.”

Fig. S9. Potential of mean force (PMF) as a function of the center-of-mass distance between two particles during sliding motions along the [010] direction (i.e., overlap increases to the left). The purple line shows the PMF obtained by sliding one particle relative to the other while maintaining a single water-layer separation in the z-direction, with both particles electrically neutral. The green line shows the PMF obtained by sliding one particle relative to the other while maintaining two water layers between the particles; in this scenario, both particles have positively charged edge surfaces with a charge density of $+0.1897$ C/m². The charge is evenly distributed on the hydrogen atoms of the edge hydroxyl groups. Cl⁻ ions are introduced to balance these positive surface charges in the simulation, ensuring overall charge neutrality.

Comment 5:

I also list some minor comments below:

- L 47 (and elsewhere): please, replace “rotational forces” by “torques”.
- L 113: replace “crossed polarized light microscopy” by “polarized-light optical microscopy” (POM).
- L 118: 10 – 20° mosaicity.
- Fig. 1f: the -3-1-1 label should be replaced by 3-1-1.
- L 164 (and elsewhere): replace “angel” by “angle”.
- L 166: replace (c,d) by (d,e).

Response and Change in Manuscript:

Following the reviewer’s suggestions, we have made the following replacements in the revised manuscript.

- L 54 and 82: “rotational forces” is replaced by “torsional forces”.
- L 113: “crossed polarized light microscopy” is replaced by “polarized-light optical microscopy”.
- L 118: “10 – 20° ” is replaced by “10 – 20° mosaicity”.
- Fig. 1f: “-3-1-1 ” is replaced by “3-1-1”.
- L166: “ (c,d)” is replaced by “(d,e)”.

Comment 6:

- L 220: The comparison with the SmC phase is not very good because this phase is a 1-dimensional stack of liquid layers. In fact, disk-like liquid crystalline molecules sometimes form columnar phases with the stacking axis at any angle with respect to the 2-dimensional lattice of columns. (See for example, A.M.Levelut et al, J. Physique, vol 42, p 147 (1981)).

Response:

We agree that the SmC phase, which involves a one-dimensional stacking of fluid layers, is not an appropriate analogy for our system, which consists of the stacking and lateral displacement of rigid, solid nanoplates. We also note that disk-like liquid crystalline molecules can adopt a variety of columnar phases with distinct orientations, as demonstrated by Levelut et al.

In response to the reviewer's insightful suggestions, we revised the manuscript to clarify these distinctions and included a citation to the work by A. M. Levelut *et al.*, as shown below:

Change in Manuscript:

Main text, Line 199-205, revised the discussion about the comparison of the liquid crystal system.

Revised discussion:

“The resulting structure resembles the columnar phase observed in gibbsite platelets (*Nature* **406**, 868-871 (2000)) and in disc-like molecules of liquid crystals (*J. Phys. France* **42**, 147-152 (1981)). However, in our system, the initial suspension of gibbsite is dilute (volume fraction = 0.81 %) compared to the high-volume fraction (> 45%, *Nature* **406**, 868-871 (2000)) typically required for liquid-crystal phases. *In situ* liquid cell TEM (LCTEM) observation of the freshly prepared suspension (i.e., without aging) reveals the presence of staggered gibbsite stacks (**Fig. S4**), indicating that mesocrystal formation arises from the self-assembly and sedimentation of nanoplates in this dilute system.”

Comment 7:

By the way, it would be interesting to determine whether or not the mesocrystals show any hkl reflection with all indices different from zero. If so, these objects are truly crystalline. If not, this means that the columnar stacks are uncorrelated and the mesocrystals can be compared to columnar liquid crystals. (See also the papers by Petukhov and Lekkerkerker on the hexagonal LC phase of gibbsite platelets)

Response:

We have re-examined our diffraction data with this specific point in mind. Regrettably, we were unable to confidently identify any (hkl) reflection with all indices non-zero. This is likely due to the absence of Bragg diffraction spots corresponding to $Mc^*(001)$, which reflects the stacking of adjacent nanoplates within

individual gibbsite columns. Our gibbsite nanoplates exhibit a thickness polydispersity of 9 ± 2 nm, which leads to variations in inter-basal spacing. This, in turn, causes broadening of the corresponding Bragg peak, merging it into the diffuse background and rendering it indistinguishable from noise. These observations suggest excellent lateral registry, but limited coherence along the stacking c-axis.

However, the SAXS diffraction spots of $Mc^*(010)$ and $Mc(01)$, together with the TEM and SEM images, reveal that the columnar stacks are correlated through 2D hexagonal stacking. Although the assembled aggregates do not exhibit three-dimensional positional order characteristic of a true single crystal, they can still be classified as mesocrystals, as the definition of a mesocrystal only requires that the constituent nanoparticles share a common crystallographic orientation at the atomic scale along at least one direction.

This scenario resembles that described by Petukhov and Lekkerkerker for the hexagonal liquid-crystalline phase of gibbsite platelets. We have now incorporated this analogy into the revised manuscript (lines 196–199) and included the relevant reference (*Nature* **406**, 868-871 (2000)), as noted in our response above.

Comment 8:

- L 265: Electrostatic interactions can be attractive, for example, in the case of ion correlations, as mentioned at the end of the sentence. Please, rephrase.

Response and Change in Manuscript:

We agree with the reviewer that electrostatic interactions can be attractive under certain conditions, and have revised the discussion accordingly:

Main text line 243-244:

Original discussion:

“Given that the electrostatic interactions can only be repulsive, the net attraction must be attributed ostensibly to a combination of ion correlation and van der Waals forces.”

Revised discussion:

“These attractions potentially arise from a combination of hydration force, ion-ion correlations, and van der Waals interactions.”

- Finally, the English style should be carefully edited in many places.

Response:

We appreciate the reviewer’s observation. To address this point, we have carried out a comprehensive language revision of the entire manuscript.

Reviewer #2:

Li et al. report a systematic study on the formation of mesocrystals of gibbsite by ordered attachment using a combination of in situ and ex-situ real-space imaging and scattering techniques as well as numerical simulations. The experiments provide a fantastically detailed characterization of the structure of the mesocrystals and make a very convincing case for this non-conventional crystal growth mechanism. They demonstrate in a – to my knowledge – unique manner the translational and orientational alignment of nanoplatelets to form mesoscopic crystals. Overall, I find the experimental work very interesting but have some concerns regarding the relevance of the numerical simulations that should be clarified.

Response:

We thank the reviewer for his/her positive evaluation of our work and for emphasizing the significance of our observations on nanoplatelet alignment during mesocrystal formation.

Main concern:

I understand that the platelets in the simulations have to be chosen smaller than in the experiments for computational reasons. While I am not so much worried about the reduced lateral dimensions, the decrease of the plate separation for 3-4nm in the experiments to 0.3nm seems rather critical. On p. 18, the authors rightfully state that ‘the barriers to sliding ... in the experiments will be greatly reduced ...’ compared to the simulations. Given the correlation length of water and the H-bonding network of about 1nm, shouldn’t one rather expect that any anisotropy is completely washed out at 3-4nm distance? If so, what is then the relevance of the simulations for the experimental observations? This must be addressed very clearly.

Response:

Upon careful re-evaluation of our *in situ* liquid-cell TEM (LCTEM) data, we acknowledge that the previously reported absolute interparticle spacing of 3–4 nm may be an overestimate due to two key factors:

1. **Image Resolution Limitations:** The raw LCTEM images (Fig.S4) have a pixel size of approximately 1 nm. As a result, any gap measurement from a single image carries an uncertainty of ± 1 nm.
2. **Imaging Artifacts from Defocus:** To enhance visibility of the low-Z, 9 nm-thick gibbsite nanoplates suspended in water (sandwiched between two SiN windows), we intentionally under-focused the objective lens of TEM. The bright “white rim” surrounding the particles is a Fresnel fringe, generated at abrupt changes in projected density or thickness. The fringe width, $w \approx (\lambda\Delta f)^{0.5}$, depends on defocus. Therefore, the presence of a white rim between the two gibbsite nanoplates suggests an abrupt change in thickness or density, implying the existence of a separation. However, the apparent 3–4 nm spacing in Fig. S4 should be considered an upper bound rather than a precise measurement.

To better assess the true interparticle distance, we collected higher-resolution LC-TEM images in ultrapure water (Fig. S5), which revealed inter-basal spacings of ~ 0.7 nm—consistent with two layers of confined water. However, precise quantification remains challenging due to projection and contrast limitations. Nonetheless, these results suggest that in many cases, the separation may be substantially smaller than previously assumed, possibly approaching 1–2 molecular water layers.

This experimentally inferred confinement regime aligns with the conditions modeled in previous molecular dynamics simulations, where we computed the potential of mean force (PMF) between two aligned gibbsite nanoplates as they approach each other along their basal surfaces (Figure 2, *J. Phys. Chem. Lett.* 2022, 13, 40, 9339–9347). The simulations reveal a deep secondary minimum (~ -5 kcal/mol \cdot nm²) for configurations with one layer of water and a shallower third local minimum (< -1 kcal/mol \cdot nm²) when separated by two water layers.

In light of the reviewer’s concern—and similar feedback from others—we recognize the importance of extending our PMF analysis to configurations that more closely resemble the experimental system. We have therefore performed new PMF calculations for particles with positively charged edges, separated by two molecular layers of water. The results, now included in Fig. S9 (also reproduced here for the reviewer’s convenience), confirm that our key conclusions are robust across both 1W and 2W scenarios:

1. The barrier during sliding motions increases with increasing overlapping area, consistent with our original findings for neutral particles separated by a single water layer (1W).
2. The completely overlapped configuration does not correspond to the global minimum for both 1W and 2W systems.
3. Hydration effects remain crucial, as demonstrated by the reduced barrier for the 2W scenario compared to the 1W scenario.
4. The PMF for the 2W case exhibits fewer distinctive features than the 1W case, likely attributable to diminished water structuring effects between the particles.

We have revised the manuscript accordingly to clarify these points and to better articulate how the simulation conditions relate to the experimental system. We appreciate the reviewer's insightful critique, which helped us to improve the rigor and clarity of our comparisons.

Change in Manuscript:

Main text, line 334-367:

“Note that in our simulation, the particle size (~ 4.2 nm in diameter) is much smaller than that in the experiments (~ 90 nm). This precludes quantitative energy barrier predictions for particle sliding in the experiments, as the relationship between the energy barrier and particle size is unknown. Additionally, the simulations assume a uniform water thickness between perfectly flat platelets, whereas TEM data (Fig. S4) show that real platelets are not perfectly flat across their entire diameter. Although the high-resolution images in the liquid cell (Fig. S5) indicate an inter-basal separation of roughly 0.7 nm—consistent with two molecular layers of water—the contrast and projection effects inherent to liquid-cell TEM prevent precise point-by-point measurement of the water-layer thickness. To better match the experiments, we perform another simulation where two particles are separated by two water layers and have positively charged edges (2W). The PMF profile (Fig. S9) indicates a reduction in energy barriers for 2W compared to 1W. This is expected, as the barriers in the PMFs arise from the hydration structure at the interface. Such barriers are known to decay rapidly as the separation increases to multiple water layers, typically becoming negligible at a separation of just three or four layers, or approximately 1 nm¹¹. Due to these discrepancies and uncertainties, the simulation results are primarily used for qualitative prediction of the free energy landscape of the particle sliding.

Finally, the PMF profiles for both neutral particles (Fig. 4) and positively charged edges (Fig. S9) provide insights into the thermodynamic origin of the uniform stagger. When sliding along the [010] direction, the completely overlapped and aligned configuration (e.g., configuration G) is less energetically favorable relative to a staggered configuration, regardless of water thickness between the two particles. This may result from a decrease in entropy due to an increase in the number of confined water molecules and the merging of two particles into one. Additionally, for the charge-

neutral gibbsite particle, the discontinuity at the edge and the under-coordinated edge-Al atoms (i.e., 5-fold coordinated Al; see Methods) introduce local charges. The staggered configuration likely mitigates electrostatic repulsion between like-charged edges. For the particles with positively charged edges, the electrostatic repulsion is stronger than that of neutral particles. This reinforces the results that the staggered configuration should dominate. The extent of particle staggering will depend on the balance among electrostatic edge-edge repulsion, van der Waals attraction (both of which depend on particle size and distance between them), and the entropy associated with sliding. Notably, in our simulations, the global energy minimum along the [010] sliding path occurs at an offset of approximately 0.7 nm from perfect overlap for small model particles (4.2 nm). When scaled to full-sized gibbsite nanoplates with a diagonal dimension of ~ 93 nm, this corresponds to a projected offset of ~ 15.5 nm, consistent with the offset distribution measured in SEM and TEM images (**Fig. S10**), which yields an average of 11.5 ± 6 nm.”

”

Fig. S9. Potential of mean force (PMF) as a function of the center-of-mass distance between two particles during sliding motions along the [010] direction (i.e., overlap increases to the left). The purple line shows the PMF obtained by sliding one particle relative to the other while maintaining a single water-layer separation in the z-direction, with both particles electrically neutral. The green line shows the PMF obtained by sliding one particle relative to the other while maintaining two water layers between the particles; in this scenario, both particles have positively charged edge surfaces with a charge density of $+0.1897$ C/m². The charge is evenly distributed on the hydrogen atoms of the edge hydroxyl groups. Cl⁻ ions are introduced to balance these positive surface charges in the simulation, ensuring overall charge neutrality.

Fig. S5. Separation of gibbsite nanoplates in a stack observed by high-resolution LC-TEM. (a, b) Low- and high-magnification TEM images of stacked gibbsite nanoplates. (c) Intensity profile across a nanoplate stack, showing periodic peaks corresponding to inter-plate separations.

Minor comments:

1. P. 13, l 255: I understand that the x-ray data show an (ensemble) average staggering of 6-12nm. The LC-TEM shows 30nm. I presume that this is based on a single observation – or is this also a statistically averaged quantity? If so, what is the error bar? If not, isn't then simply the lack of statistics an equally likely explanation as the proposed hindered diffusion due to the walls?

Response:

The estimation of the average staggering of 6-12 nm is derived from the TEM and SEM observations. We also calculate the stagger distance based on the SAXS data as follows:

The stagger distance extracted from the (01) reflections can be written as:

$$\langle s \rangle = t \tan \theta = 12.2 \pm 2.7 \text{ nm},$$

where $t = 9.5 \pm 2.1$ nm is the platelet thickness (obtained from TEM analysis), and θ is the monoclinic tilt angle. Because the uncertainty in θ is difficult to quantify—due to challenges in performing reliable background subtraction in the SAXS patterns—we report only the mean value here.

We also re-measured the lateral offset directly using SEM and conventional TEM, which yielded an average value of 11.5 nm with a standard deviation of 6.0 nm, and a maximum observed value of 28 nm (Fig. S10).

The previously cited trajectory (offset ≈ 30 nm) is now supplemented by a second movie (Movie S2 and Fig. S6), which shows a platelet pair halting at an offset of 18 nm. Considering both movies together ($n = 2$), the observed offsets are 18 nm and 30 nm—approximately +1 s.d. and +3 s.d. relative to the ex-situ mean, thus well within the long-tailed distribution.

We agree that limited sampling alone could account for offsets larger than the mean. Accordingly, we have revised our original statement to explicitly acknowledge statistical variability as a contributing factor.

Change in Manuscript:

Main text, lines 232-235:

Original discussion:

“Throughout the movement, the average sliding rate decreases (Fig. 3f) and the sliding stops at the configuration where the centers of two nanoplates are offset by about 30 nm along the [010] direction. This offset is higher than the average offset measured for the mesocrystal (6-12 nm), which may reflect a limitation on the ability of the nanoplates to diffuse within the thin water layer and/or the interaction between the membrane and the nanoplates.”

Revised discussion:

“The final offsets recorded *in situ* (18–30 nm, Fig. S5) exceed the ensemble mean (≈ 12 nm) but fall within the upper tail of the *ex situ* distribution (Fig. S10). This suggests that both statistical variability and the restricted Brownian mobility within the liquid-cell gap may contribute.”

2. Do I understand correctly that the platelets are forced to move strictly along the (010) and (100) direction without allowing neither lateral nor vertical displacements of the center-of-mass nor orientational fluctuations? Isn't that very somewhat unrealistic, in particular if the distance between platelets is 3-4nm? I would expect a substantial reduction of the relevant barriers if such fluctuations would be included.

Response:

We thank the reviewer for raising this important point. In response to the main concern (Comment 1), we have revised the manuscript to clarify that the interparticle separation is approximately 0.7 nm—not 3–4 nm as previously estimated—based on improved analysis of our *in situ* LC-TEM data. This thinner confinement regime supports the relevance of the constrained motion used in our simulations.

To further address the reviewer's concerns and suggestions from other reviewers, we performed additional *in situ* LC-TEM experiments under identical experimental conditions. The new dataset (Movie S2, recorded at 1 frame/s) captures a gibbsite nanoplate approaching and attaching to another platelet in a manner fully consistent with our earlier observations (Movie S1). However, in Movie S2, the nanoplate moves more slowly, allowing us to resolve its trajectory with greater precision.

The center-of-mass trajectory after the nanoplate undergoes flipping ($t = 11.0$ s) is shown in Fig. S6g. The path reveals a sliding motion initially along the [010] direction, followed by movement toward the crystallographically equivalent $[-130]$ direction. Minor “random-walk”-like deviations are visible, especially as the projected overlap increases, but the overall drift remains strongly aligned with the [010]/ $[-130]$ axes when averaged over tens of nanometers.

These new measurements confirm that the sliding is not strictly constrained but rather guided by an anisotropic free energy landscape. While short-term fluctuations are present, the long-range motion preferentially follows crystallographically defined directions.

Regarding vertical (out-of-plane) motion, we acknowledge that our current LC-TEM setup cannot resolve such fluctuations due to projection limitation. This possibility remains open and could be further investigated in future studies. However, in other OA studies involving nanoparticles, vertical fluctuations at the contact interface are frequent. Consistent with this, our PMF results show that increasing the interparticle separation (from one to two water layers) reduces the sliding barrier, reflecting the effect of decreased confinement. Given that we currently lack direct evidence for their occurrence in our system, we have not included this possibility in the comparison between simulation and experiment.

Change in the manuscript:

Figure S6 has been added in the SI.

Fig. S6. In situ liquid-cell TEM visualization of the oriented attachment (OA) of gibbsite nanoplates. (a–f) Time-lapse TEM images from Movie S2 of two nanoplates as they approach and attach. White dots mark the moving nanoplate’s center of mass projected onto the imaging plane, and white lines connect its current position of the center-of-mass to the earlier positions. (g) Detailed trajectory of the center of mass of the attaching nanoplate after it flips into partial alignment ($t = 11.0$ s). During sliding, the nanoplate changes its direction of motion from the [010] axis to the [1-30] axis. The [010] and [1-30] directions are crystallographically equivalent. (h) Average sliding speed of the attaching nanoplate versus time, showing that its velocity gradually decreases as time passes, and the overlap area increases.

3. P. 19, line 392: ‘The contrast ... highlights the role of hydration barriers ...’: again, I don’t see the evidence for this beyond the simulations, which are in a different regime than the experiments. (see main concern above)

Response:

We believe that our response to the reviewer’s main concern (see above) adequately addresses this important issue.

4. P. 5, line 104: AFM measurements reported attractive forces between the gibbsite basal plane and (negatively charged) silica tips below pH 6 indicating positive zeta potentials and surface charges (see Gan&Franks, Langmuir 2006; Klaassen et al. Nanoscale 2017)

Response:

We thank the reviewer for pointing this out. We have addressed the comment by citing the suggested references in the revised introduction, to better clarify the surface charge characteristics of the gibbsite basal plane.

Change in Manuscript:

Main text, lines 107-112:

“Recent AFM studies have revealed that even the (001) basal surface carries a slight but measurable positive surface charge at $\text{pH} < 6$ ²⁷, primarily due to ion adsorption at defect sites²⁸. Despite this, the basal plane remains weakly charged relative to the edge surfaces, which carry $\equiv\text{Al}(\text{OH}_2)^{2+}$

groups with pKa values of 9.0–10.0. As a result, the nanoplates can be viewed as having quasi-neutral basal surfaces and positively charged edges under our experimental conditions (pH 5.6, low ionic strength).”

5. P. 11, line 207: does the ‘direction rather than positional order’ mean that the d-spacing is not well-defined? (I may misunderstand something here.)

Response:

By “orientational, but not full positional, order,” we mean that the gibbsite nanoplates are aligned along a common stacking direction (Gb[001]), as supported by SAXS anisotropy and WAXS indexing. However, the spacing between adjacent columns is not uniform enough to yield a well-defined d-spacing. This is evidenced by the absence of sharp $Mc^*(001)$ or $Mc^*(011)$ Bragg reflections, and instead the presence of diffuse equatorial streaks in the SAXS pattern, indicative of short-range correlations and a distribution of inter-platelet spacings.

This lack of translational coherence likely arises from the thickness polydispersity of the nanoplates (9 ± 2 nm), which broadens and weakens the stacking reflections. These observations imply good lateral registry but limited periodicity along the stacking (c) axis. We have revised the manuscript to clarify this distinction.

Change in Manuscript:

Main text, lines 191-193:

“The absence of $Mc^*(001)$ reflections indicates orientational alignment of gibbsite nanoplates along the Gb[001] direction, without clear long-range translational periodicity.”

6. Typos:

p. 13. Line 269: ‘basal planes’

p. 6, line 128: ‘... [100] and [100] ...’ ???

p. 7, caption Fig. 1: captions speaks of ‘the mesocrystal’, the text of ‘a mesocrystal’. Do all images show the same or just ‘a’ mesocrystal?

p. 7: Fig. 1c: dark blue labels are invisible when printed on paper

p. 9: Fig. 2 cation : ‘...wide/small angel ...’ or angle ;-)

Response:

We thank the reviewer for carefully identifying these typos and inconsistencies. We have corrected all of them in the revised manuscript.

Reviewer #3:

Response:

We are grateful for the time and effort invested by all reviewers in improving our manuscript.

Reviewer #4:

Comment 1:

The manuscript by Xiaoxu Li et al presents a remarkable experimental work on the formation and structure of a mesocrystal of charged gibbsite particles in deionized water. The work combined WAXS/SAXS/SEM/TEM/LCTEM to carefully elucidate the resulting crystalline ordered mesostructure. The manuscript is also very well written, which does not detract from the rest.

Response:

We sincerely thank the reviewer for the generous and encouraging comments.

Comment 2:

Finally, the manuscript presents simulations and relies on them to try to explain the formation of the observed mesocrystal. It is on this last point that my problem is concentrated. The simulations are based

on the observation (and not the other way around) that the particles slide over each other and attempt to show, in a way that I find unconvincing, why they stop sliding before perfect overlap, thus giving rise to the observed angle between the normal to the hexagonally ordered columns and the normal to the basal plane of the platelets. Guided by these observations, the authors chose to use molecular simulations to measure the free energy landscape of interaction between two particles of gibbsite forced to slide in relation to each other in two crystallographic directions while maintaining a constant basal distance. As might be expected, the free energy profile shows oscillations alternating between attraction and repulsion, which increase with the overlap of the particles. The authors conclude that perfect overlap is not achieved for kinetic reasons, that is particles stop sliding because the free energy barrier increases with the particle overlap. But this explanation contradicts another remark made in the manuscript: “The uniformity of the lattice parameters suggests that the staggered arrangement of platelets represents a thermodynamic minimum in the degree of overlap”. The authors have to choose: either the structure is kinetically arrested or it is thermodynamically stable, but it cannot be both at the same time.

Response:

We thank the reviewer for pointing out this important issue. We acknowledge that the statement suggesting particles stop sliding due to increasing free energy barriers is misleading and does not reflect our intended interpretation. This statement appeared in both the abstract and the final paragraph of the manuscript (main text, lines 381–391), and we agree that it contradicts our central conclusion.

To clarify: throughout this work, our interpretation has been that the staggered configuration represents a **thermodynamically stable state**, as supported by the uniform lattice parameters observed in the mesocrystal and by our PMF simulations. The simulations show that perfect overlap does not correspond to the global free energy minimum. Instead, the lowest free energy occurs at a finite, staggered offset between particles. This arrangement reduces the repulsive interactions associated with high overlap and allows for more favorable water-mediated interactions between surfaces.

Moreover, our LC-TEM experiments provide direct evidence of preferential sliding along the [010] direction, with the motion decelerating as the overlapping area increases. To further understand this behavior, our MD simulations were designed to explore the underlying free energy landscape and to address two additional questions: (1) why sliding is energetically favored along specific crystallographic directions, and (2) how energy barriers evolve with increasing overlap—thereby elucidating the mechanism behind the observed slowdown in motion. The simulation results are consistent with the experimental observations of directional preference and decreasing mobility with increasing overlap. They also provide new insight into the critical role of interfacial water in achieving long-range ordered mesostructures, as further supported by comparison with the case of boehmite.

In light of this, we have revised the manuscript to convey our interpretation that the observed uniform staggered structure has a **thermodynamic origin**.

Change in the manuscript:

Abstract, lines 26-29:

Original text:

“Molecular dynamics simulations confirm that sliding along the [010] direction is energetically favorable, and that the energy barrier rises with increasing overlap until correct attachment is achieved.”

Revised text:

“Molecular dynamics simulations reveal that this staggered arrangement corresponds to a global free energy minimum, rather than full alignment. The simulations also confirm that sliding along the [010] direction is energetically favored and provide insight into the role of interfacial water in achieving long-range ordered assemblies.”

Main text, lines 391-393:

Original text:

“Due to the increase in the energy barrier during sliding, the kinetics decrease over time, and the nanoplate sliding stops at a partially overlapped configuration, resulting in the offset of nanoplates along the [010] direction observed within the mesocrystal.”

Revised text:

“The balance among electrostatic edge-edge repulsion, van der Waals attraction (both of which depend on particle size and distance between them), hydration force, and the entropy associated with sliding leads to the formation of the staggered gibbsite mesocrystals.”

Comment 3:

Beyond the semantics, the simulations carried out have a number of flaws: the positive electrostatic charges at the edge of the particles and the many-body interactions in the mesostructure are ignored. However, the repulsive electrostatic interactions coupled with the many-body interactions could very well explain the staggered arrangement of the particles, see for example the article by Delhorme et al (ref 16). As Helmut Coelfen pointed out, “A temporary stabilization of the nanoparticles is crucial for the nanoparticle orientation to a mesocrystal by interaction forces to maintain that the system is able to find the point of minimal energy on its energy landscape” (DOI: 10.1002/adma.200901365). Following the same logic, it would have been appropriate to try to determine the minimum free energy of interaction between the two particles. Incidentally, it can be noted that the minimum free energy of the two calculated profiles provided in the manuscript does not correspond to a perfect particle overlap (for the simulated system presented, this would result in an angle of about forty degrees if my calculations are correct).

Response:

We thank the reviewer for this thoughtful and detailed comment. In response to the reviewer's concern regarding electrostatic edge charges, we have conducted an additional set of simulations in which the particle edges carry a surface charge density of $+0.1897 \text{ C/m}^2$, consistent with known values for gibbsite under experimental conditions. As shown in Fig. S9, these simulations confirm that the global minimum in the potential of mean force (PMF) still occurs at a finite, staggered overlap, and not at perfect alignment. Importantly, the inclusion of edge charges does not alter the key thermodynamic conclusions of our study.

Regarding many-body effects, we agree that interactions beyond the pairwise level may influence long-range ordering in real mesocrystals. However, our focus here is on two-particle interactions, which represent the fundamental building blocks of the stacking process. The simplified pairwise model enables tractable calculation of the PMF while still capturing key features such as hydration-mediated barriers and electrostatic contributions. We acknowledge that future extensions to many-body simulations would provide a more complete picture of mesostructure formation.

We appreciate the reviewer's reference to the work of Delhorme et al. (ref. 16), which considered negatively charged basal surfaces. In contrast, our gibbsite system has neutral basal planes, and its energetics are dominated by hydration interactions at those surfaces. Since the total area of the basal surfaces is significantly larger than that of the edges, and because experimental evidence indicates minimal net charge on the basal planes under our conditions, our simulations prioritize water-mediated interactions, particularly hydrogen bonding, between neutral surfaces.

We also fully agree with Helmut Cölfen's view that the ability of particles to maintain a metastable separation is crucial to navigating the energy landscape toward mesocrystal formation. In fact, this is the essential point of our simulations, which support this idea by showing that particles separated by a single or double water layer can reach a thermodynamically favorable staggered configuration via lateral sliding, even without a full collapse into direct contact. We clarify that our simulations are not designed to capture the final "jump to contact" process, but rather to identify stable configurations along the interaction pathway.

Regarding the reviewer's observation that the free energy minimum does not correspond to perfect overlap—and may imply an angular offset—we agree. In fact, our simulations consistently show that perfect overlap is not thermodynamically favored, regardless of the sliding direction ([100] or [010]), edge charge (neutral or positive), or interparticle separation (1W or 2W). The staggered arrangement that emerges from these simulations correlates well with the angular offset observed experimentally between the column axis and the basal planes.

Regarding the reviewer's estimate of a $\sim 40^\circ$ angle based on the simulated configuration, we acknowledge that this provides an interesting geometric interpretation. We have clarified this point in the revised manuscript to better connect the simulation findings with the experimentally observed mesostructure.

To better communicate these points, we have revised the relevant section of the manuscript to emphasize the thermodynamic nature of the simulation results, as copied below:

Main text, line 334-367:

“Note that in our simulation, the particle size (~ 4.2 nm in diameter) is much smaller than that in the experiments (~ 90 nm). This precludes quantitative energy barrier predictions for particle sliding in the experiments, as the relationship between the energy barrier and particle size is unknown. Additionally, the simulations assume a uniform water thickness between perfectly flat platelets, whereas TEM data (**Fig. S4**) show that real platelets are not perfectly flat across their entire diameter. Although the high-resolution images in the liquid cell (**Fig. S5**) indicate an inter-basal separation of roughly 0.7 nm—consistent with two molecular layers of water—the contrast and projection effects inherent to liquid-cell TEM prevent precise point-by-point measurement of the water-layer thickness. To better match the experiments, we perform another simulation where two particles are separated by two water layers and have positively charged edges (2W). The PMF profile (**Fig. S9**) indicates a reduction in energy barriers for 2W compared to 1W. This is expected, as the barriers in the PMFs arise from the hydration structure at the interface. Such barriers are known to decay rapidly as the separation increases to multiple water layers, typically becoming negligible at a separation of just three or four layers, or approximately 1 nm¹¹. Due to these discrepancies and uncertainties, the simulation results are primarily used for qualitative prediction of the free energy landscape of the particle sliding.

Finally, the PMF profiles for both neutral particles (Fig. 4) and positively charged edges (Fig. S9) provide insights into the thermodynamic origin of the uniform stagger. When sliding along the [010] direction, the completely overlapped and aligned configuration (e.g., configuration G) is less energetically favorable relative to a staggered configuration, regardless of water thickness between the two particles. This may result from a decrease in entropy due to an increase in the number of confined water molecules and the merging of two particles into one. Additionally, for the charge-neutral gibbsite particle, the discontinuity at the edge and the under-coordinated edge-Al atoms (i.e., 5-fold coordinated Al; see Methods) introduce local charges. The staggered configuration likely mitigates electrostatic repulsion between like-charged edges. For the particles with positively charged edges, the electrostatic repulsion is stronger than that of neutral particles. This reinforces the results that the staggered configuration should dominate. The extent of particle staggering will depend on the balance among electrostatic edge-edge repulsion, van der Waals attraction (both of which depend on particle size and distance between them), and the entropy associated with sliding.

Notably, in our simulations, the global energy minimum along the [010] sliding path occurs at an offset of approximately 0.7 nm from perfect overlap for small model particles (4.2 nm). When scaled to full-sized gibbsite nanoplates with a diagonal dimension of ~ 93 nm, this corresponds to a projected offset of ~ 15.5 nm, consistent with the offset distribution measured in SEM and TEM images (**Fig. S10**), which yields an average of 11.5 ± 6 nm.”

Fig. S9. Potential of mean force (PMF) as a function of the center-of-mass distance between two particles during sliding motions along the [010] direction (i.e., overlap increases to the left). The purple line shows the PMF obtained by sliding one particle relative to the other while maintaining a single water-layer separation in the z-direction, with both particles electrically neutral. The green line shows the PMF obtained by sliding one particle relative to the other while maintaining two water layers between the particles; in this scenario, both particles have positively charged edge surfaces with a charge density of $+0.1897$ C/m². The charge is evenly distributed on the hydrogen atoms of the edge hydroxyl groups. Cl⁻ ions are introduced to balance these positive surface charges in the simulation, ensuring overall charge neutrality.

Comment 4:

Finally, the authors forget the role of the gravitational field. Although it is negligible for the simulated system considered, it is one of the main drivers of the sedimentation of gibbsite particles and therefore of the formation of the observed mesocrystals. On the same subject, the authors may wish to read the work of Lekkerkerker and in particular these two articles (DOI: 10.1140/epjst/e2013-02075-x; <https://pubs.acs.org/doi/10.1021/la100797x>) observing the role of gravity on the formation of hexagonal columnar phase where average platelet orientation is decoupled from the column axis.

To conclude I find the experimental study and its results very interesting, but the conclusions drawn from the simulations are not convincing. At the very least, the latter need to be moderated and discussed, particularly with regard to the numerous approximations (neglected physics) and hypotheses on which they are based.

We acknowledge the limitation of our simulations in neglecting gravitational effects. This point is now addressed in the manuscript as follows:

Main text, lines 368-375:

“Gravity, which is neglected in our simulations, can shift this balance for sufficiently large platelets. Van der Heijden et al. showed that gravity compacts columns formed by large gibbsite platelets (≈ 570 nm in diameter), leading to denser packing, pronounced column undulations, and a misalignment between platelet orientation and the column axis near the bottom of the structure³⁹. In contrast, our single-aggregate SAXS patterns for 90 nm platelets (**Fig. 2e**) reveal a uniform, coherent tilt of the hexagonal columns with no orientational fluctuations. This consistency suggests that at the sub-100 nm length scale, van der Waals, electrostatic, and hydration forces dominate over gravitational torque, preserving columnar alignment.”

Point-by-Point Response

General improvement

The authors appreciate the valuable comments from all four reviewers. Below, we provide detailed replies to each of the referees' comments (shown in *italic blue*) along with the changes made in our revised manuscript. The corresponding line numbers in the revised manuscript, in which we have tracked the changes, are also indicated. When substantial modifications were necessary, we included the new text below for ease of assessment. We hope that the revised version of our manuscript is now suitable for publication in Nature Communications.

Reviewer #1 (Remarks to the Author):

The authors have revised their manuscript according to my comments. In particular, they have reproduced the in-situ TEM experiments and shortened the X-ray scattering section. They have also addressed my minor comments. I am not familiar enough with molecular simulations to properly assess this part, but I believe that the in-situ TEM study, on its own, is novel and exciting enough to justify publication of this article in Nature Communications. Therefore, I recommend publication of this revised manuscript with no further changes.

Response:

We sincerely thank the reviewer for their positive evaluation and recommendation for publication.

Reviewer #2 (Remarks to the Author):

I appreciate the efforts of the authors regarding the repeated experiments and the attempt to reconcile the experimental spacing of the basal planes and the one used in the simulations, as well as the charge along the edges.

Response:

We thank the reviewer for acknowledging our efforts.

I watched the new (and the old) TEM movie over and over again in an attempt to convince myself whether it is fair to draw far-reaching conclusions based on these data. My honest opinion is not quite. I understand that recording these data is a great achievement as such – and I do also see that both videos clearly demonstrate a process in which an initially vertically standing platelet lies down flat and subsequently moves more or less in plane. However, whether that motion is along specific crystallographic axes or not, cannot be extracted from this very limited data set in convincing manner. The original movie simply contains three snapshots only, implying two consecutive steps in the same direction. The new one, upon sliding the movie back and forth from $t=11s$ to approx. $33s$ shows more a kind of rotation around the top corner than a directed motion along any specific crystallographic axis. In both cases, there are also adjacent particles to the bottom left (old movie) and on the bottom right (new movie) with which the moving particle seems to interact/collide. Based on the data presented, this might just as well be the cause for the change of direction as the suggested anisotropic hydration structure of the water in between the platelets. The case for the

intrinsic directional diffusion would be more convincing if there were no neighbouring particles to interact with.

Response:

We sincerely appreciate the reviewer's careful examination and comments on the LCTEM movies, which helps us improve the manuscript.

In response, we re-analyzed the same movie frame-by-frame with a more systematic and transparent pipeline as shown in revised Fig. 3, focusing on the specific points raised.

We quantified both translational and rotational motion of the platelet with higher temporal precision and explicit error estimation. The particle contours in each frame were extracted using ImageJ. The translational displacement centroid positions were re-calculated relative to the initial frame according to detailed contours. Instantaneous speed was obtained via a 3-point central difference method, which minimizes random frame-to-frame noise. Rotations were quantified using the Kabsch algorithm, which computes the optimal rigid-body rotation between successive contours, with uncertainties derived from bootstrapping the contour coordinates.

Fig. 3a show selected frames illustrating the transition from an initially tilted configuration to an in-plane arrangement. In Fig. 3b, the centroid trajectory after flip ($t = 11.0$ s) is color-coded by time, with the crystallographic [100] and [010] directions of attached gibbsite aggregates indicated for reference. Several neighboring particles are outlined with dashed lines. Fig. 3c shows the displacement (blue), instantaneous speed (green), and rotation (red) over time, each with corresponding error bars. The sustained net translation follows the crystallographically defined [010] direction, while rotational changes are comparatively small (within $\sim 10^\circ$) and do not correlate with transient neighbor proximity, indicating that the motion is intrinsic and directionally biased rather than dominated by random collisions.

From 11 s onward, there is a relatively rapid rotation leading to a maximum $\sim 10^\circ$ misalignment, which is subsequently corrected over time. This $\sim 10^\circ$ rotation is also consistent with our previous PMF calculations of basal-basal interaction forces as a function of rotation, as well as with the selected-area electron diffraction results from TEM. As shown in Fig. S4, the basal surfaces of gibbsite nanoplates are not perfectly flat, leading to local variations in interplate spacing. These topographic features, together with minor surface asperities, can induce small in-plane rotations during sliding without altering the overall directional preference.

We have now clarified these points in the section "***The OA of gibbsite nanoparticles by sliding***" to explicitly state that the observed transient rotation does not contradict the overall conclusion of preferential sliding along Gb[010], and that alternative explanations such as collisions have been carefully evaluated and excluded based on the data.

Changes in the manuscript:

Main text lines 225-251:

"The *in situ* observations reveal a two-stage OA process: (i) rotation and jump-to-contact to achieve partial basal-basal overlap, and (ii) sliding primarily along the Gb[010] direction to maximize overlap toward a staggered configuration. After several seconds of electron beam irradiation, edge-attached gibbsite nanoplate detached from the membrane, rotated, and flipped to a position of partial overlap with the closest gibbsite nanoplate via basal-basal attachment within ~ 1 s. The initial detachment is likely due to the accumulation of positive charges on the Si_xN_y surface from

secondary electron emission³⁴. The following motion prioritizes the alignment of nanoplates on their basal surfaces over maximizing the overlapping area, indicating the presence of strong, long-range attractive forces between their basal surfaces. These attractions potentially arise from a combination of hydration forces, ion-ion correlations, and van der Waals interactions³⁵.

After the flip at ~ 11 s, the nanoplate—initially well aligned with the underlying one—underwent an in-plane rotation of $\sim 10^\circ$ (Fig. 3b), likely caused by a local perturbation during the onset of sliding (e.g., minor surface asperities as shown in Fig. S4). No collisions with neighboring particles were observed during this period. The misalignment was gradually corrected, consistent with PMF predictions²⁹ of a strong preference for basal–basal alignment within $\sim 10^\circ$ and with SAED confirming the final alignment (Fig. S2).

During stage (ii), the nanoplate slid mainly along the Gb[010] direction (Fig. 3b), increasing overlap while progressively slowing down (Fig. 3c). This sliding direction matches the tilting direction of nanoplate columns in mesocrystals, indicating a preferred pathway for alignment. Similar two-stage behavior was consistently observed (Movie S2), although sliding rates varied between events. These differences likely arise from variations in hydrodynamic drag due to local liquid confinement: changes in the water-layer thickness between plates can strongly influence viscous resistance and particle mobility, as reported for nanoparticle diffusion in thin water films at solid–liquid interfaces^{34,36}. The final offsets recorded in situ (18–30 nm, Figs. 3 and S6) exceed the ensemble mean (≈ 12 nm) but fall within the upper tail of the ex situ distribution (Fig. S10), suggesting contributions from statistical variability and restricted Brownian mobility in the liquid-cell gap.”

Detailed data processing was added in **Methods** session (lines 450-455):

“The particle contours in each frame were extracted using ImageJ42. The translational displacement centroid positions were calculated relative to the initial frame according to detailed contours. Instantaneous speed was obtained via a 3-point central difference method, which minimizes random frame-to-frame noise. Rotations were quantified using the Kabsch algorithm, which computes the optimal rigid-body rotation between successive contours, with uncertainties derived from bootstrapping the contour coordinates.”

Fig. 3 has been revised based on original Fig. S6:

Fig. 3. In situ liquid cell TEM observation of orientated attachment of gibbsite nanoplates. (a) Selected frames (2.2 s/frame) showing the transition from an initially tilted platelet to an in-plane configuration. (b) Outlines and centroid trajectory of moving nanoplate after flip ($t = 11.0$ s). Several neighboring particles are outlined with dashed lines. (c) Displacement (blue), instantaneous speed (green), and rotation (red) of the moving nanoplate over time, extracted from the centroid trajectory in panel (b).

I understand that it is extremely difficult to carry out and interpret these experiments. Regarding my original main concern, the on-edge views of the particle stacks indeed show that spacings much thinner than the original estimate of 3-4nm do indeed occur with values that are compatible with two monolayers of H₂O. This is an important addition. I understand that it is impossible to extract the actual distances from the top view TEM images in Figs. 3 and S6.

Response:

We truly appreciate the reviewer's acknowledgment of this key clarification.

While the authors state that the new data set in Fig. S6 is 'fully consistent with our original observation', the translation speed reported in Fig. S6h is 100x slower than in Fig. 3f for a very similar size of the particles. It does not become clear to me why this is 'fully consistent'. (Possibly, the separation for the old data is indeed 3-4nm as originally suggested whereas for the new ones it is indeed much thinner leading to stronger friction – but this is all speculative.)

Response:

We acknowledge that the sliding speed in original Fig. S6h (now in Fig. 3c) is markedly lower than in another movie, despite the comparable lateral size of the nanoplates. Both events occur within the confined

geometry of the liquid cell, but the degree of local confinement may differ. In the latter uploaded dataset, the moving nanoplate is positioned above at least two underlying particles, which could further limit the available liquid space between the plates and the cell windows. Together with the intrinsic surface roughness of the gibbsite nanoplates (Fig. S4), this may lead to a thinner and more structured interplate water layer, increasing viscous resistance. In contrast, the earlier dataset may have involved a thicker water layer and fewer confinement constraints, potentially allowing faster sliding.

Previous in situ studies have shown that the diffusion coefficient of nanoparticles in thin aqueous films at solid–liquid interfaces can be reduced by 4–5 orders of magnitude compared with bulk liquid, due to increased viscosity and structural ordering of interfacial water layers as the liquid gap narrows (Environ. Sci. Technol. 2016, 50, 11, 5606–5613; Langmuir 2015, 31, 25, 6956–6964). For a Brownian-like sliding process, the characteristic speed scales approximately with the square root of the diffusion coefficient. Thus, a 4–5 order-of-magnitude decrease in diffusion coefficient would be expected to lower the sliding speed by ~100–300 times, matching the difference observed between Fig. 3 and Fig. S6.

We have clarified this in the revised manuscript as follows, noting that small differences in interplate water-layer thickness and local confinement between events can markedly affect hydrodynamic drag and hence the observed sliding rates, without altering the underlying mechanism.

Lines 244-248:

“Similar two-stage behavior was consistently observed (Movie S2), although sliding rates varied between events. These differences likely arise from variations in hydrodynamic drag due to local liquid confinement: changes in the water-layer thickness between plates can strongly influence viscous resistance and particle mobility, as reported for nanoparticle diffusion in thin water films at solid–liquid interfaces^{34,36}.”

Regarding Fig. 3f, I also don't understand the first data point at $t=2.4$ s. If I understand correctly, this data point originates from Δr_1 in Fig. 3b. However, that first step of the center of mass translation involves not only a translation but also the re-orientation process, which probably makes up for an important fraction of Δr_1 . It is therefore not justified to treat this data point on the same footage as the two subsequent ones. With only two data points left, however, the statement about the gradually decreasing lateral sliding speed with increasing particle becomes a bit futile.

Response:

We agree that comparing only two data points is not sufficient to support the statement on gradual deceleration. In the revised manuscript, Fig. 3c now presents the detailed instantaneous speed of the nanoplate as a function of time, which clearly shows a slowdown as the overlapping area increases. The original Fig. 3 has been moved to Fig. S6, and we have revised its caption to clarify that the first data point ($t \approx 2.4$ s) corresponds to Δr_1 in panel (b) and contains both translational and rotational components during the flip–contact step.

Changes in the manuscript:

Fig. S6. *In situ* liquid-cell TEM observation of gibbsite nanoplate OA from Movie S2. (a–d) Time-lapse TEM images; scale bar: 100 nm. (e) Schematic illustration of the OA process and relative motions of the nanoplates. (f) Average 2D projected translational velocity ($\Delta r/\Delta t$) of the moving nanoplate during OA. The first data point ($t \approx 2.4$ s) corresponds to Δr_1 in panel (b) and includes both translational and rotational contributions during the flip–contact step; therefore, it is not directly comparable to the subsequent two points, which represent primarily lateral sliding. Nevertheless, the latter two points clearly show a decrease in sliding speed with increasing overlap area, consistent with Fig. 3.

I am also a bit confused about this new statement: ‘Notably, in our simulations, the global energy minimum along the [010] sliding path occurs at an offset of approximately 0.7 nm from perfect overlap for small model particles (4.2 nm). When scaled to full-sized gibbsite nanoplates with a diagonal dimension of ~93 nm, this corresponds to a projected offset of ~15.5 nm, consistent with the offset distribution measured in SEM and TEM images (Fig. S10), which yields an average of 11.5 ± 6 nm. Assuming that the water layer thickness is indeed 0.7 nm, the suggested scaling would imply that the staggering angle scales with the particle size. If, however, the staggering angle is caused by the electrostatic edge-to-edge repulsion, why would this scale with the particle size?’

Response:

We agree that the scaling from 0.7 nm (simulation, ~4.2 nm diagonal) to ~15.5 nm (experimental size, ~93 nm diagonal) assumes that the relative offset distance—expressed as a fraction of the total particle diagonal—remains approximately constant across sizes. As discussed in the manuscript, the extent of staggering arises from a balance among electrostatic edge–edge repulsion, van der Waals attraction, and the entropy of sliding. However, our PMF calculations (Fig. S9) show that the position of the energy minimum along [010] changes only slightly—from 0.70 nm for uncharged particles in monolayer water to 0.75 nm for charged particles in bilayer water—indicating that electrostatic effects make only a minor contribution. The dominant factor is the local hydration structure at the basal–basal interface, which is independent of the overall particle size. Under this assumption, the constant fractional offset observed in simulations naturally scales linearly with the particle diagonal, yielding ~15.5 nm for experimental-sized gibbsite nanoplates.

Taking everything together and looking at the raw data, I don't feel that the amount and detail of LC-TEM data presented – notwithstanding its beauty – is sufficient to support the conclusion of a strongly directional and hydration-mediated lateral translation mode.

Response:

We believe our detailed responses to the comments above adequately address the reviewers' concerns.

Reviewer #3 (Remarks to the Author):

Response:

We thank both reviewers for their time, effort, and constructive evaluation of our work.

Reviewer #4 (Remarks to the Author):

The authors have responded satisfactorily to my comments. I have no further comments to make.

Response:

We greatly appreciate the positive feedback and the recognition of our responses.